



# The decline of alpine lichen heaths generates atmospheric heating but subsurface cooling during the growing season

Peter Aartsma[1*], Johan Asplund[2], Arvid Odland[1], Stefanie Reinhardt[1], Hans Renssen[1]

[1]Department of Natural Sciences and Environmental Health, University of South-Eastern Norway, Gullbringvegen 36, NO-3800, Bø, Norway
[2]Faculty of Environmental Sciences and Natural Resource Management, Norwegian University of Life Sciences, P.O. Box 5003, NO-1432, Ås, Norway

*Correspondence to:* Peter Aartsma (peter.aartsma@usn.no)

**Abstract.** Lichen heaths are declining in abundance in alpine and arctic areas partly due to an increasing competition with shrubs. This shift in vegetation types might have important consequences for the microclimate and climate on a larger scale. The aim of our study is to measure the difference in microclimatic conditions between lichen heaths and shrub vegetation during the growing season. With a paired plot design, we measured the net radiation, soil heat flux, soil temperature, and soil moisture on an alpine mountain area in south Norway during the summer of 2018 and 2019. We determined that the daily net radiation of lichens was on average 3.15 MJ (26%) lower than for shrubs during the growing season. This was mainly due to a higher albedo of the lichen heaths, but also due to a larger longwave radiation loss. Subsequently, we estimate that a shift from a lichen heath to shrub vegetation leads to an average increase in atmospheric heating of 3.35 MJ per day during the growing season. Surprisingly, the soil heat flux and soil temperature were higher below lichens than below shrubs during days with high air temperatures. This implies that the relatively high albedo of lichens does not lead to a cooler soil compared to shrubs during the growing season. We hypothesize that the thicker litter layer, the presence of soil shading, and a higher evapotranspiration rate at shrub vegetation are far more important factors in explaining the variation in soil temperature between lichens and shrubs. Our study shows that a shift from lichen heaths to shrub vegetation in alpine and arctic areas will lead to atmospheric heating, but has a cooling effect on the subsurface during the growing season, especially when air temperatures are relatively high.



## 1 Introduction

Lichen heaths are one of the most dominant vegetation types across alpine and arctic areas (Cornelissen et al., 2001). For example, lichen heaths cover up to 6% of Norway (Bryn et al., 2018). Besides their extensive abundance, lichens are important forage for reindeer during winter (Heggberget et al., 2002; Vistnes and Nellemann, 2008). However, the lichen cover has

decreased in alpine and arctic areas during the last decades (Cornelissen et al., 2001; Joly et al., 2009; Elmendorf et al., 2012; Lang et al., 2012; Fraser et al., 2014; Maliniemi et al., 2018). For instance, Fraser et al. (2014) estimated that lichen cover decreased by 24% in the western Canadian Arctic between 1980 and 2013. Also, large continuous lichen mats are rarely observed anymore in this region, while they were common 40 years ago (Fraser et al., 2014). Similar declining trends have been observed throughout the alpine and arctic areas. The lichen decline is attributed to an increased competition with vascular

plants that benefit from climate change, especially shrubs (Cornelissen et al., 2001; Fraser et al., 2014; Moffat et al., 2016; Vuorinen et al., 2017; Chagnon and Boudreau, 2019). Experimental warming studies show that this lichen decline has the potential to proceed with the ongoing temperature increase (Walker et al., 2006; Elmendorf et al., 2012). Therefore, it is important to study the consequences of the lichen decline on alpine and arctic ecosystems.

Shrubs benefit from recent climate change, since the higher temperatures and longer growing seasons are in favor of
their growing conditions (Myers-Smith et al., 2011; Myers-Smith et al., 2020). Indeed, many studies found an increase in shrub cover, biomass and abundance in alpine and arctic areas over the past decades (Sturm et al., 2001b; Hallinger et al., 2010; see Myers-Smith et al. (2011) for a review). Such an increase of shrubs alters the vegetation composition in these areas (Pajunen et al., 2011; Boscutti et al., 2018). For example, multiple studies have reported a negative relationship between shrubs and lichen occurrence (Cornelissen et al., 2001; Pajunen et al., 2011; Maliniemi et al., 2018). Moreover, Chagnon and Boudreau

(2019) found a lower lichen abundance and diversity below shrubs compared to areas without shrubs. These studies imply that shrub vegetation outcompetes the lichens heaths in the long run. This might alter the alpine and arctic environment in various ways, since lichens and shrubs have distinct characteristics. For example, Aartsma et al. (2020) measured an average albedo of 0.255 for lichen heaths, while the average albedo of shrubs was 0.132. Therefore the shift from lichen-dominated areas to shrub-dominated areas might have, among others, important consequences for the microclimate and the large-scale climate of

alpine and arctic areas.

Extensive studies have shown that shrub expansion has a substantial impact on microclimatic conditions, including surface albedo, soil temperature and permafrost stability (Myers-Smith et al., 2011; Loranty et al., 2018). Chapin et al. (2005) estimated that a shift from tundra to a complete shrub environment has the potential to increase the atmospheric heating substantially. Contrary to this increase in atmospheric warming, shrubs can have a cooling effect on the subsurface due to

shading by the canopy. Myers-Smith and Hik (2013) found that summer soil temperatures were 2 °C lower below a shrub cover than below shrub-free patches due to shading of the soil by the shrub canopy. The shading effect also reduces permafrost thaw below shrubs (Blok et al., 2010). However, it is expected that the large scale increase in atmospheric heating due to shrub expansion will overwhelm the cooling effect of shading and soil temperature will increase below shrubs in the long-term





(Lawrence and Swenson, 2011; Bonfils et al., 2012). Therefore, the general accepted view is that shrub expansion has a positive
feedback on climate warming (Pearson et al., 2013), although, some uncertainties still exist (Loranty and Goetz, 2012).

While we have a good understanding of how shrubs affect the climate, the impact of lichens on the micro- and large-scale climate has not been studied in a thorough way yet. It is anticipated that the lichens' insulating properties and their high albedo will have a cooling effect on the micro- and large-scale climate (Bernier et al., 2011; Porada et al., 2016). For example, Odland et al. (2017) found a negative correlation between lichen abundance and soil temperature on Norwegian mountain
summits. Also, Porada et al. (2016) modelled the impact of lichens and bryophytes on the soil temperature at high latitudes. They estimated that lichens and bryophytes lower the soil temperature on average by 2.7 °C compared to an environment without lichens and bryophytes. However, they considered only the insulating properties of the two vegetation types and not the high albedo of lichens. Therefore, lichens might decrease the soil temperature even more. Most of the field measurements on the influence of lichens on the subsurface microclimate are based on differences between lichens and bare soil or disturbed
lichens (e.g. Fauria et al., 2008; Nystuen et al., 2019; Van Zuijlen et al., 2020). However, constructive field measurements on the difference in soil temperature between lichens and shrubs are lacking and therefore it is uncertain how the observed shift from lichen-dominated areas to shrub-dominated areas will change the micro- and large-scale climate in alpine and arctic areas.

To address this issue, we have set up a study to measure the difference in microclimatic conditions between lichen heaths and shrub vegetation at a mountain site in Norway. Our study design follows recommendations to apply a vegetation-
specific approach to come to more detailed conclusions on the impact of shrub expansion and lichen decline (Stoy et al., 2012; Juszak et al., 2016; Williamson et al., 2016; Loranty et al., 2018). We focus on four microclimatic variables: net radiation, soil heat flux, soil temperature, and soil moisture. We used a paired plot design to measure these variables simultaneously at lichen and shrub plots in a Norwegian mountain area during two summers. Due to the paired plot design, we ensured that the lichen and paired shrub plots face similar background weather conditions, topographical characteristics and parent material. We
hypothesize that lichens will have a lower net radiation because of their high albedo. Due to this lower net radiation and the insulating properties of lichens (Porada et al., 2016), we further hypothesize that the soil heat flux and soil temperature will be lower under lichens than under shrubs. With this study, we advance the knowledge on the impact of lichens on the microclimate during the growing season, which is important to answer the question how a future vegetation shift from lichen-dominated areas towards shrub-rich environments might alter the micro- and large-scale climate.

## 2 Methods

### 2.1 Study area

The study was conducted at Imingfjell (60.1901° N, 8.5724° E), a mountain area in southern Norway with an elevation ranging from 1100 to 1350 m a.s.l. The vegetation is typical low alpine zone vegetation. Windswept ridgetops are covered with lichen heaths (see Appendix A for a picture of the area). Most common lichen species are of the genera *Cladonia*, *Flavocetraria*,
*Alectoria* and *Cetraria* (Aartsma et al., 2020). The most abundant shrub species in the area is *Betula nana*, mainly located on





the midslope and ridgetop positions. The parent material of the soils in the study area consists of metarhyolitic moraine material (NGU, 2020). The nearest weather station (Dagali, 25 km from the study site; 828 m a.s.l., MET Norway (2019), Station nr. 29790) reported an average yearly temperature of 0.5 °C with an average July temperature of 11 °C for the period 1988–2007. The average yearly precipitation during this period was 550 mm.

**2.2 Data collection**

We selected a study site of 2.5 km along a county road and 200 meters from this road into the field, resulting in an area of ca. 50 ha. In this study site, we delineated the lichen heaths using areal images of Geonorge (2018) in ArcMap (ESRI, 2019). The delineated lichen patches had a total area of 15 ha. Within these patches, we randomly selected ten locations. Subsequently, we used the criteria in Table 1 to select a lichen-dominated plot and a shrub-dominated plot near each of these ten locations

(Fig. 1). We measured the climatic variables simultaneously and in an identical way in one of the paired lichen and shrub plot at the time for two days. After these two days, we moved the sensors to the next paired plots. We conducted the measurements on these plots between 4 July and 13 August 2018. Days with a precipitation duration of more than 30 min were excluded to minimize the effect of precipitation on the radiation measurements.

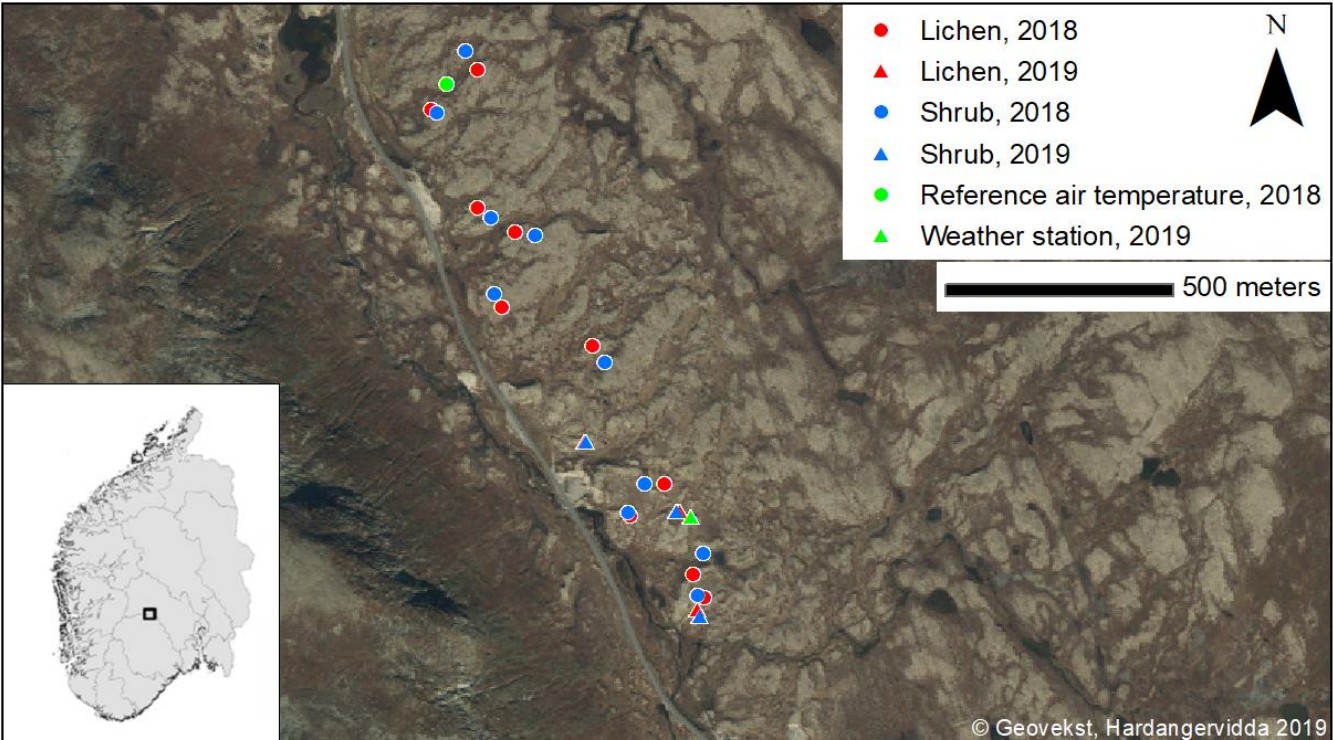

**Figure 1: Locations of the plots in the study area. Inset: location of Imingfjell in southern Norway.**




**Table 1: Criteria for the selection of lichen and corresponding shrub-dominated plots. Table from Aartsma et al. (2020).**

| Plot characteristics | Criteria |
|---|---|
| Location lichen plot | $\leq 50$ meters from random location |
| Location shrub plot | $\leq 50$ meters from selected lichen plot |
| Landscape position | Ridgetop |
| Vegetation composition | $\geq 50\%$ of targeted vegetation type |
| Non-vegetation surface | $\leq 10\%$ |
| Size | $\geq 4\ m^2$ |
| Radius | $\geq 112\ cm$ |
| Slope angle | $\leq 10°$ |
| Aspect | $\leq 10°$ difference between paired plots |
| Other | Undisturbed (e.g. no grazing) |
| | No overlap with already chosen plot |

The terms of the net radiation, the soil heat flux, the soil temperature and the soil moisture were measured in a similar way on similar positions in each plot (Fig. 2). We measured the incoming shortwave radiation, reflected shortwave radiation,

incoming longwave radiation and outgoing longwave radiation in W m$^{-2}$ with one Kipp & Zonen CNR4 net radiometer per plot. We placed the radiometer 30 cm above the canopy, which led to a measurement radius of 112 cm. With this height, we ensured that all the measured reflected shortwave radiation was reflected by the studied plot. The radiometer measured every 5 s and the data loggers (Kipp & Zonen Logbox SE) collected 5 min averages. We measured the soil heat flux at two positions per plot with Hukseflux HFP01SC self-calibrating heat flux sensors. We placed the heat flux sensors at 5 cm depth below the

soil surface and measurements were done every 5 min. These measurements were recorded with Campbell Scientific CR800 data loggers. We measured the soil temperature on three positions per plot and at each of these positions on two depths (1 and 5 cm below the soil surface) with LogTag TRIX-8 temperature loggers. The temperature loggers measured the soil temperature every 5 min. We measured soil moisture at the same three positions as the soil temperature with ECH$_2$O 5TM soil moisture sensors at 5 cm below the soil surface. These sensors measured the soil moisture every 5 min and the measurements were

recorded with Em50 data loggers. We measured the reference air temperature at 1 m height at one location in the study area (Fig. 1) with an UTL-3 Temperature Datalogger placed in a Stevenson screen throughout the field season. In addition, we measured the precipitation manually with a regular rain gauge.

We measured the vegetation height in every plot at 10 cm intervals along a North–South and an East–West transect. This led to 49 height measurements per plot. The thickness of the litter layer was determined at each of the five positions where

soil temperature or soil heat flux were measured in each plot. We drilled one hole in the middle of each plot with a soil auger (Ø 4 cm) and described the soil using the FAO guidelines (WRB, 2006). In each plot, we took three soil samples of the upper



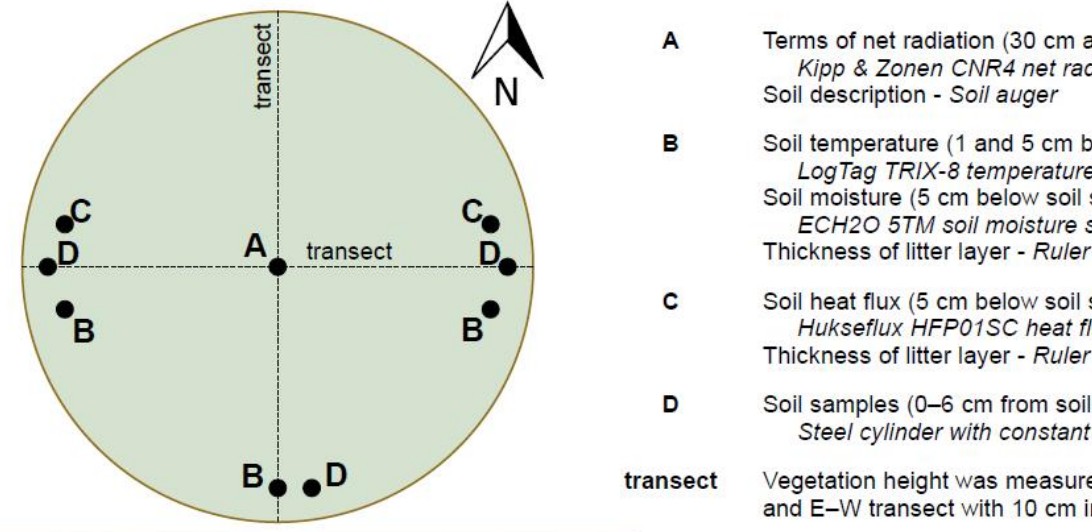

| | |
|---|---|
| **A** | Terms of net radiation (30 cm above canopy)<br>*Kipp & Zonen CNR4 net radiometer*<br>Soil description - *Soil auger* |
| **B** | Soil temperature (1 and 5 cm below soil surface)<br>*LogTag TRIX-8 temperature logger*<br>Soil moisture (5 cm below soil surface)<br>*ECH2O 5TM soil moisture sensor*<br>Thickness of litter layer - *Ruler* |
| **C** | Soil heat flux (5 cm below soil surface)<br>*Hukseflux HFP01SC heat flux plates*<br>Thickness of litter layer - *Ruler* |
| **D** | Soil samples (0–6 cm from soil surface)<br>*Steel cylinder with constant volume* |
| **transect** | Vegetation height was measured along the N–S<br>and E–W transect with 10 cm interval - *Ruler* |

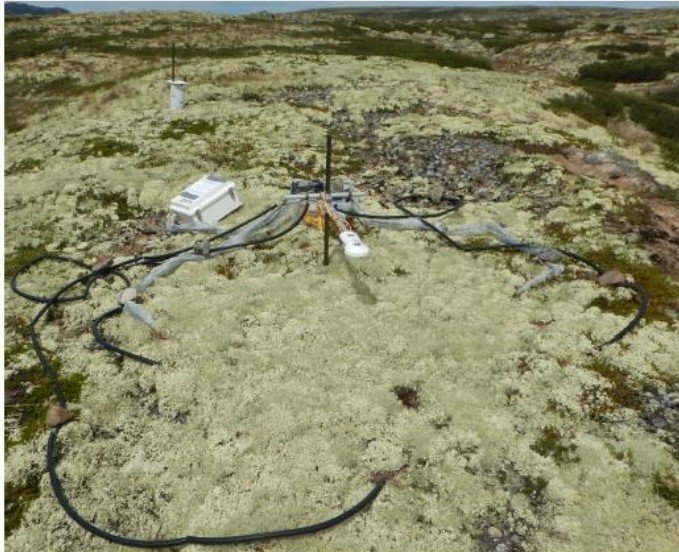

**Figure 2: A schematic overview of the positions of the measurements within one plot, a picture of the setup and a list of measured variables per plot. The sensors and materials are indicated in *italic*. All variables were measured at the same positions within all lichen and shrub plots.**

6 cm of the mineral soil by filling a metal cylinder (volume 265 cm$^3$) twice per sample. If the metal cylinder encountered a rock during insertion into the soil, we took a new sample in close vicinity. The two sub-samples were mixed and dried at 105 °C for 24 h. Subsequently we weighed the samples and calculated the bulk density per sample. In addition, we measured the

organic matter percentage using the loss on ignition method (Heiri et al., 2001, ignition conditions: 550 °C for 3 h) and measured the particle size distribution by dry sieving using an Endecott E.F.L. 1 MK11 sieve shaker.

The two-day measurements performed in 2018 were complemented in 2019 with measurements for 6 subsequent days in three additional paired lichen and shrub plots. In contrast to the plots of 2018, we selected the locations of the paired plots of 2019 subjectively (Fig. 1). However, the plots of 2019 also fulfilled the criteria of Table 1. We measured one paired plot at

the end of June, one paired plot at the end of July and one paired plot in mid-August. To monitor the background weather





conditions in a more thorough way than during the 2018 field season, we placed a HOBO RX3000 remote weather station at the study site for the 2019 field season (see Appendix B for a list of sensors of the weather station).

## 2.3 Microclimate calculations

For each plot, we calculated the net radiation (Q*) with the four terms of the radiation balance using Eq. (1) (Oke, 2002):

$$Q^* = SW_{in} - SW_{out} + LW_{in} - LW_{out} \qquad (1)$$

in which $SW_{in}$ is the incoming shortwave radiation, $SW_{out}$ is the reflected shortwave radiation, $LW_{in}$ is the incoming longwave radiation, and $LW_{out}$ is the outgoing longwave radiation.

We corrected the measurements of the soil heat flux for heat storage above the heat flux plates using Eq. (2) (Oke, 2002):

$$Q_{G0} = Q_{Gz} + C_s * (\Delta T / \Delta t) * z \qquad (2)$$

in which $Q_{G0}$ is the soil heat flux at the soil surface, $Q_{Gz}$ is the measured soil heat flux at depth z, $C_S$ is the heat capacity of the soil above the plate and $\Delta T / \Delta t$ is the change in temperature of the soil above the plate. For this correction, we converted the 5 min measurements to hourly averages and used the soil temperature that was measured at 1 cm depth to calculate $\Delta T / \Delta t$. We determined $C_s$ using Eq. (3) (De Vries, 1963):

$$C_s = 1.92 \times x_{min} + 2.50 \times x_{org} + 4.18 \times x_w \qquad (3)$$

in which $x_{min}$, $x_{org}$ and $x_w$ are the volume fractions of the mineral soil, organic matter and water, respectively. We obtained the volume fractions of the mineral soil and organic matter with the organic matter and bulk density measurements. At two plots, the $Q_{G0}$ could not be calculated at one position due to the loss of a $x_{org}$ measurement and malfunctioning of a soil temperature sensor at 1 cm depth.

## 2.4 Data analysis

For every plot of the field season of 2018, we calculated an average vegetation height, thickness of the litter layer, percentage of soil organic matter, bulk density, and silt- and clay fraction. To test if these variables and the thickness of the A-horizon (obtained during the soil description) differ significantly between the lichen and shrub plots, we performed a Wilcoxon signed rank test.

To test for differences in microclimatic conditions between the lichen and shrub plots of 2018, we used linear mixed
models for the net radiation, soil heat flux, soil temperature, and soil moisture. In the mixed models, we utilized vegetation type (lichen or shrub) and the reference air temperature with interaction as fixed effects and day of measurement nested into plot number as random effect to account for the paired sample design. We added the reference air temperature as fixed effect since we expected that it affects the response variables directly (soil temperature and soil heat flux) or indirectly by being a proxy for the general weather conditions (net radiation and soil moisture). Per microclimatic variable, we constructed separate
models for daily measurements, daytime measurements (08:00–22:00 LT) and nighttime measurements (22:00–8:00 LT). Therefore, we converted the 5 min measurements of the net radiation and hourly measurements of $Q_{G0}$ into daily, daytime and



nighttime totals (in megajoule) and the 5 min measurements of soil temperature and soil moisture into daily, daytime and nighttime averages. Only the soil temperature measured at 5 cm depth was used for this analysis.

The longer period of measurements per paired plot during the field season of 2019 allowed us to study the difference
in microclimatic conditions between lichens and shrubs over a longer time period. We constructed time series of the hourly averages of the reference air temperature, net radiation, soil heat flux and soil temperature for the three paired plots to gain more insight in the specific dynamics of the variables. In addition, we analyzed the difference in microclimatic conditions between lichens and shrubs during a warm, sunny day and a cold, cloudy day. As a basis for this analysis, we selected from one paired plot measurements from a distinct warm, sunny day and measurements from a distinct cold, cloudy day, and
constructed time series of the reference air temperature, net radiation, soil heat flux and soil temperature.

All statistical analyses were made using R version 4.0.2 (R Core Team, 2020). The mixed models were constructed with the package *nlme* (Pinheiro et al., 2011).

## 3 Results

### 3.1 Canopy and soil variables

We found a significant difference in vegetation height, thickness of litter layer and thickness of A-horizon between the lichen and shrub plots (Fig. 3). Almost no plant litter was present under the lichen plots, while we measured an average ($\pm$ SE) thickness of 7.1 ($\pm$ 0.2) cm under the shrub plots. We found no significant difference in soil organic matter, bulk density, and silt and clay fraction between the lichen and shrub plots. Moreover, there was no clear difference in soil type between the lichen and shrub plots. All soils were classified as podzols or showed clear signs of podzolisation.

### 3.2 Microclimatic conditions throughout the field season

The daily total net radiation, daily total soil heat flux and daily average soil temperature differed significantly between the lichen and shrub plots of 2018 (Table 2, Fig. 4, Fig. 5). The shrub plots had a higher net radiation than the lichen plots during the entire field season (Fig. 4b). This difference in net radiation was mainly initiated by a difference in SW* ($SW_{in} - SW_{out}$) between the vegetation types (Fig. 6), governed by the higher albedo of the lichens compared to the shrubs, since $SW_{in}$ values
were the same. On average, the daily net radiation was 3.15 MJ (26%) lower for the lichen plots than for the shrub plots. The daily total soil heat flux and daily average soil temperature were higher under lichens than under shrubs for a substantial amount of days during the field season (Fig. 4c,d) and this difference was significant when air temperatures are relatively high (Fig. 5b,c). There was no significant difference in soil moisture between the lichen and shrub plots (Table 2, Fig. 4e, Fig. 5d). The measurements of the three paired lichen and shrub plots of 2019 showed similar patterns to 2018 for the four microclimatic
variables (Appendix C).



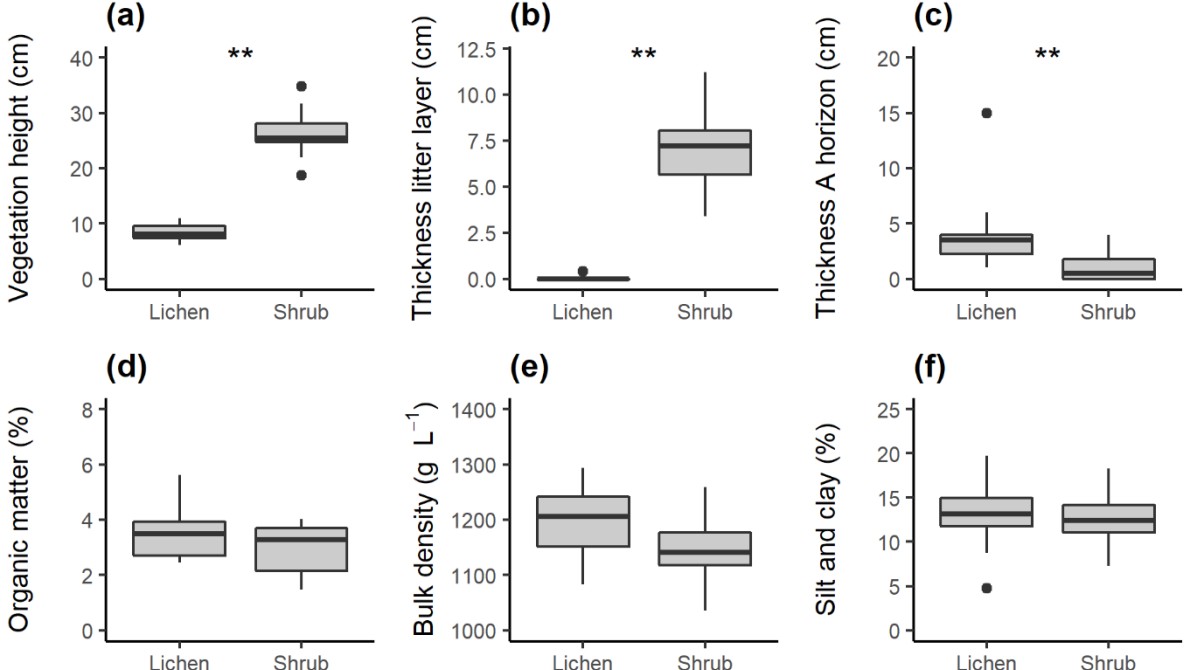

**Figure 3: Difference in vegetation height (a), thickness of the litter layer (b), thickness of the A-horizon (c), organic matter (d), bulk density (e), and silt and clay fraction (f) between the lichen and shrub plots. Asterisks indicate the significance according to the Wilcoxon signed rank test. "**" = p-value < 0.01, " " = no significant difference.**

**Table 2: Results of the linear mixed models for the effect of vegetation type on the microclimatic conditions. Different models were constructed for the daily averages/total, the daytime averages/totals (08:00–22:00 LT) and the nighttime averages/totals (22:00–8:00 LT). Statistical significant effects (p<0.05) are indicated in bold.**

| Variable | Fixed effect | Daily | | Daytime | | Nighttime | |
|---|---|---|---|---|---|---|---|
| | | F-value | p-value | F-value | p-value | F-value | p-value |
| Net Radiation | Vegetation | 0.22 | 0.646 | 0.41 | 0.531 | 3.30 | 0.086 |
| | Ref. air temperature | **7.17** | **0.025** | **10.09** | **0.011** | 0.45 | 0.517 |
| | Vegetation x ref. air temp. | **13.82** | **0.002** | **16.13** | **< 0.001** | **5.69** | **0.028** |
| Soil heat flux | Vegetation | **5.22** | **0.026** | 3.14 | 0.082 | **14.45** | **< 0.001** |
| | Ref. air temperature | **46.33** | **< 0.001** | **43.92** | **< 0.001** | **30.73** | **< 0.001** |
| | Vegetation x ref. air temp. | **9.23** | **0.004** | **9.92** | **0.003** | **4.86** | **0.032** |
| Soil temperature | Vegetation | **28.91** | **< 0.001** | **15.74** | **< 0.001** | **32.92** | **< 0.001** |
| | Ref. air temperature | **31.69** | **< 0.001** | **28.01** | **< 0.001** | **32.63** | **< 0.001** |
| | Vegetation x ref. air temp. | **53.06** | **< 0.001** | **34.81** | **< 0.001** | **54.04** | **< 0.001** |
| Soil moisture | Vegetation | 0.25 | 0.619 | 0.88 | 0.350 | 1.25 | 0.267 |
| | Ref. air temperature | 3.62 | 0.090 | 3.34 | 0.101 | 1.10 | 0.322 |
| | Vegetation x ref. air temp. | 1.96 | 0.165 | 3.41 | 0.068 | 0.00 | 0.990 |





**Figure 4: Daily measurements during the field season of 2018 for the background weather conditions (air temperature and precipitation) (a), net radiation (b), soil heat flux (c), soil temperature (d), and soil moisture (e). Every day is divided into the lichen and corresponding shrub plot for graphs (b), (c), (d), and (e). For ease of display, we calculated averages per plot for the soil heat flux (n=2), soil temperature (n=3) and soil moisture (n=3). Error bars display minimum and maximum values for soil heat flux and standard errors for soil temperature and soil moisture. Vertical dotted lines indicate a plot change.**




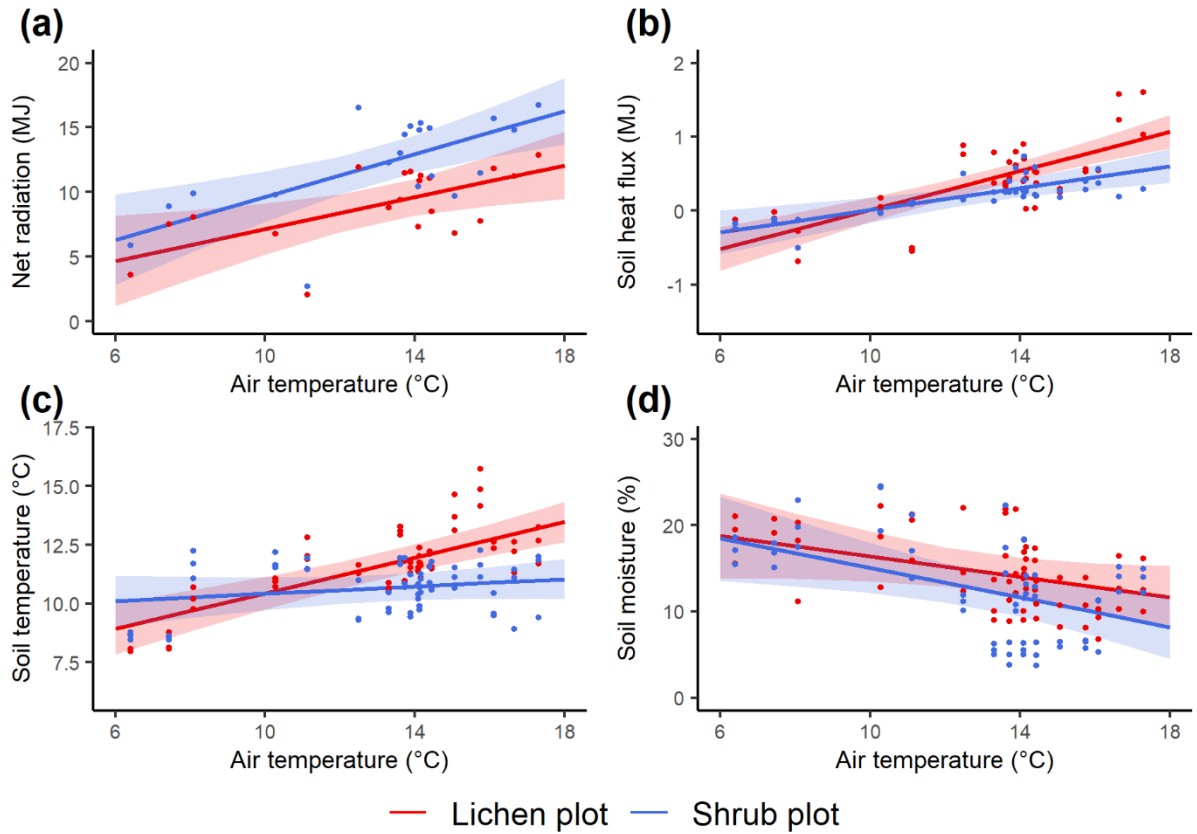

**Figure 5: The modelled relationships with 95% confidence intervals between vegetation type, daily average reference air temperature and daily total net radiation (a), daily total soil heat flux (b), daily average soil temperature (c), and daily average soil moisture (d). Points denote the actual measurements.**

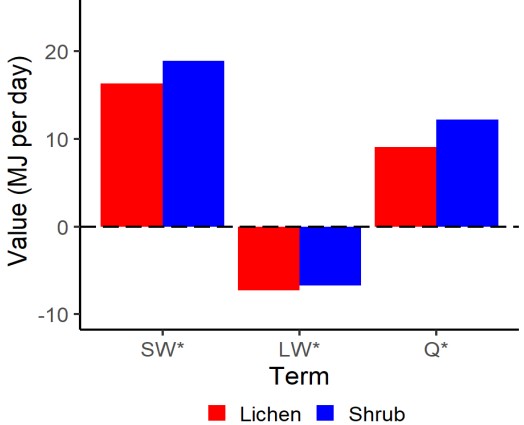

**Figure 6: Average radiation balance for all lichen and shrub plots of the entire field season of 2018. SW\* = SW$_{in}$ − SW$_{out}$ and LW\* = LW$_{in}$ − LW$_{out}$.**



### 3.3 Day vs night

The difference in daily total net radiation between the lichen and shrub plot arose during daytime (Fig. 7a, Fig. 8b). The higher
       albedo of lichens compared to shrubs will have its effect on the net radiation only during the day due to the absence of
       shortwave radiation at night. The soil heat flux below lichens was larger than below shrubs during daytime, while it was smaller
       or even negative below lichens during nighttime (Fig. 7b, Fig. 8c). As a consequence, the daily amplitude of the soil heat flux
       was larger for a lichen plot than for a shrub plot (Fig. 8c). The daily amplitude of the soil temperature was also larger for
lichens than for shrubs (Fig. 8d), but the soil temperature differed only significantly between lichens and shrubs during daytime
       with higher air temperatures for our measurements of 2018 (Fig. 7c).

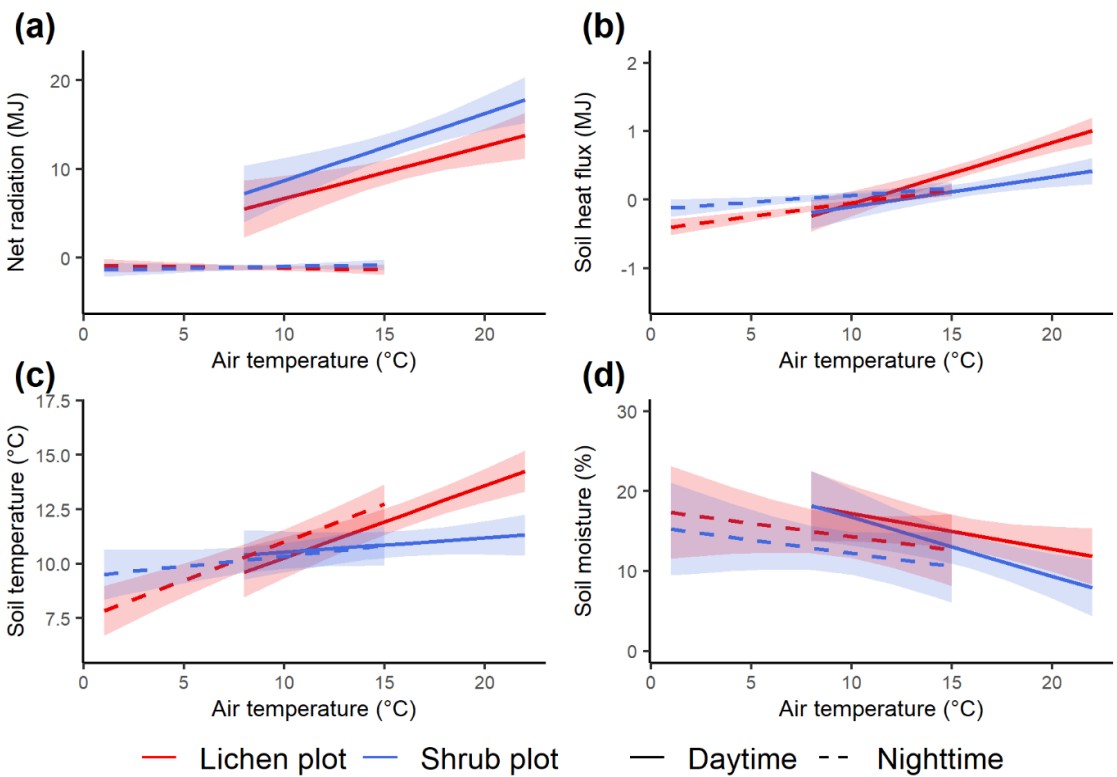

**Figure 7: The modelled relationships with 95% confidence intervals between vegetation type, average reference air temperature and total net radiation (a), total soil heat flux (b), average soil temperature (c), and average soil moisture (d) during daytime and nighttime.**



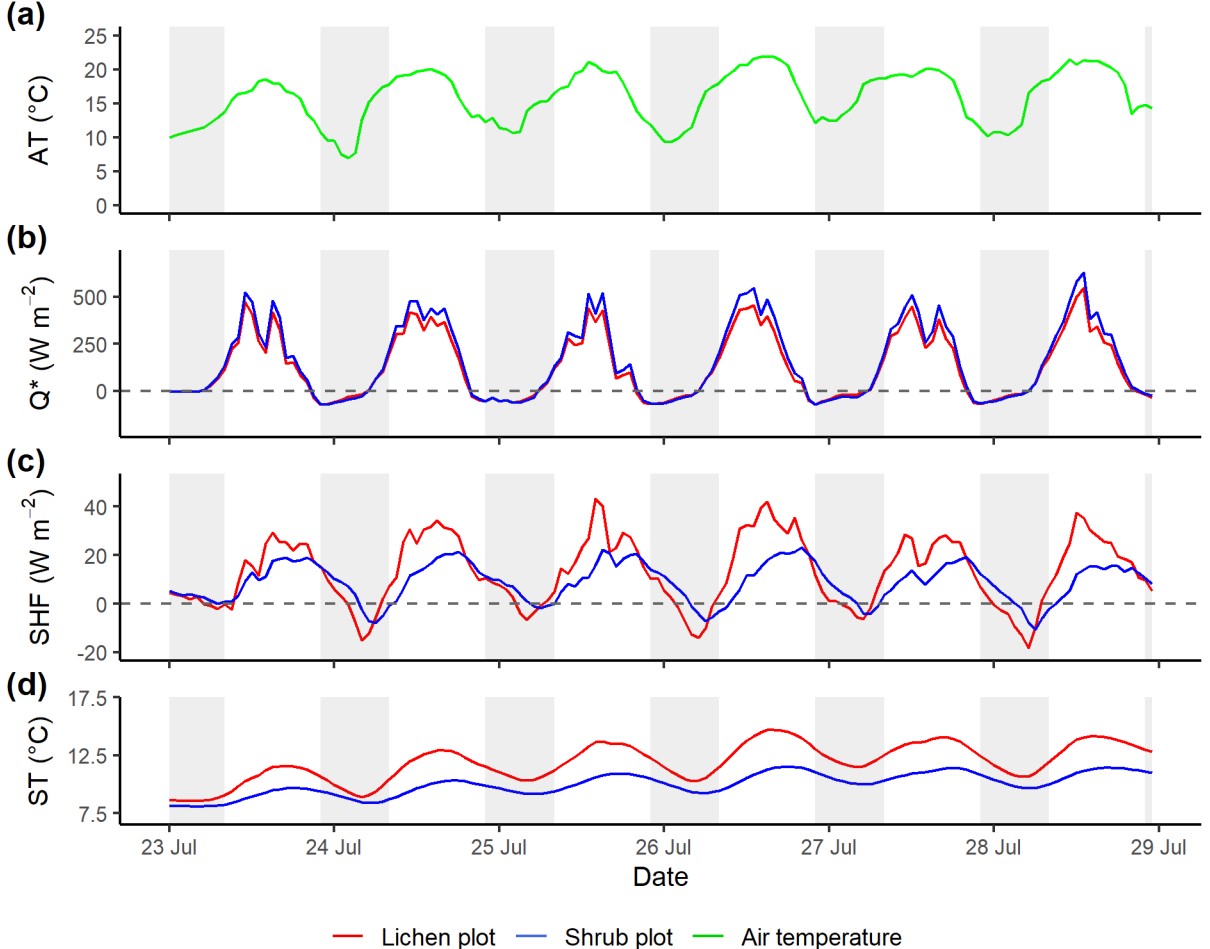

**Figure 8: The reference air temperature (a) and difference in net radiation (b), soil heat flux (c) and soil temperature (d) between one lichen and paired shrub plot during six days of the field season of 2019. Time series are constructed with hourly averages. Shaded areas indicate nighttime (22:00–8:00 LT). AT = reference air temperature, Q\* = net radiation, SHF = soil heat flux, ST = soil temperature. See Appendix D for time series of the other two paired plots of 2019.**

## 3.4 Warm and sunny day vs cold and cloudy day

The difference in the microclimatic variables between the lichen and shrub plots were more pronounced during a warm, sunny day than during a cold, cloudy day (Fig. 9). The difference in net radiation was larger during a sunny day, since the incoming
shortwave radiation is relatively high and therefore the higher albedo of lichens played a more dominant role in the net radiation. As for the net radiation, the difference in soil heat flux between the lichen and shrub plot was larger during a warm, sunny day. However, the soil heat flux was higher below the lichen plot than below the shrub plot.

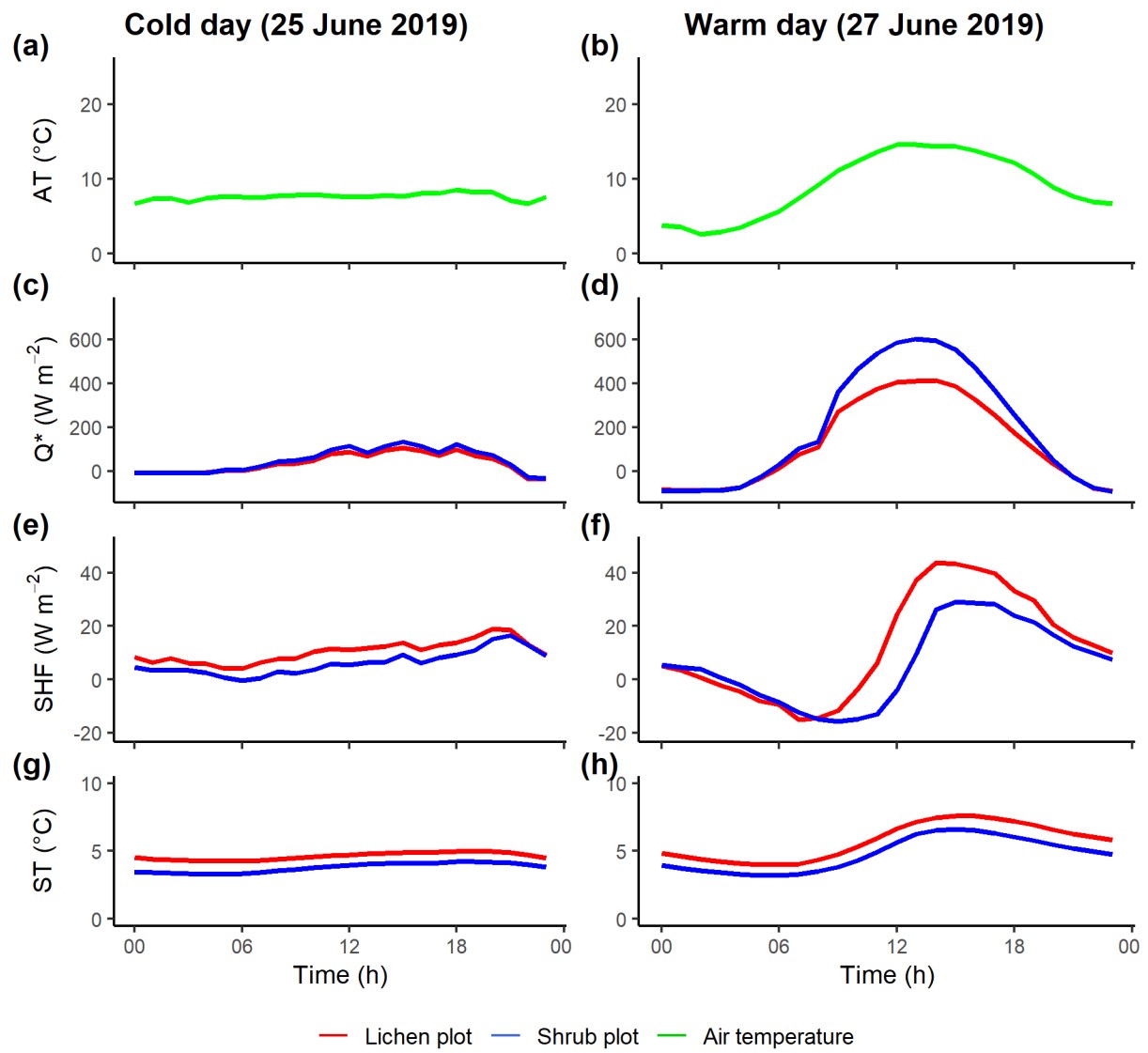

**Figure 9: The reference air temperature (a,b) and difference in net radiation (c,d), soil heat flux (e,f) and soil temperature (g,h) between one lichen and paired shrub plot during a cold, cloudy day and a warm, sunny day. AT = reference air temperature, Q\* = net radiation, SHF = soil heat flux, and ST = soil temperature. The same paired plots are used for both days.**

## 4. Discussion

### 4.1 Radiation balance

The higher net radiation of the shrub plots compared to the lichen plots was in line with our hypothesis. This difference is mainly initiated by the higher albedo of the lichen plots, as SW\* is higher for shrubs than for lichens while the difference in





LW* (LW$_{in}$ – LW$_{out}$) is marginal (Fig. 6). Moreover, previously we measured an average difference in albedo of 0.124 between the lichen and shrub plots of 2018 (Aartsma et al., 2020). The daily average net radiation of our shrub plots was 3.15 MJ higher

than of our lichen plots. Chapin et al. (2005) reported an increase in atmospheric heating of 0.55 MJ per day when the alpine tundra shifts into shrubs. Using their definition of atmospheric heating (sensible + latent heat flux, i.e. net radiation – soil heat flux), we measured an average difference in atmospheric heating of 3.35 MJ per day between our lichen and shrub plots. This difference is more than six times larger than estimated by the study of Chapin et al. (2005). However, Chapin et al. (2005) assumed that the albedo of shrubs is 0.02 higher than the albedo of alpine tundra, which is substantially lower than the

difference in albedo between our lichen and shrub plots. Moreover, our value of atmospheric heating might be slightly overestimated, since our measurements were conducted during a relatively warm and sunny field season. This is reflected among others in the relatively large daily mean SW$_{in}$ that we measured during our field season (255 W m$^{-2}$) compared to long term studies at similar latitudes (200 W m$^{-2}$, Eugster et al., 2000). Nevertheless, our study shows that large variations in radiation dynamics exist within alpine tundra depending on the vegetation composition.

The marginally lower LW* for the lichen plots (Fig. 6) is surprising, since it implies that the surface of lichens is warmer than the surface of shrubs. The larger longwave radiation loss of the lichen plots is a result of a larger LW$_{out}$, since LW$_{in}$ is similar for the paired lichen and shrub plots. Due to the dependence of LW$_{out}$ on the surface temperature following Stefan-Boltzmann's law (Oke, 2002), a larger LW$_{out}$ for the lichen plots suggests a higher surface temperature for the lichen plots, which seems counterintuitive considering the higher albedo of lichens. Moreover, time series of the LW$_{out}$ show that the

difference in LW$_{out}$ between lichens and shrubs is made during daytime, while there is no difference during nighttime (Appendix E). This points to additional processes that dominate over the effect of the albedo, showing an opposite effect. Contrasting and counterintuitive results have also been found by previous studies. For example, Stoy et al. (2012) measured a higher surface temperature for the lichen species *Cladonia rangiferina* than for the moss species *Sphagnum fuscum* despite the higher albedo of *C. rangiferina*, while Gauslaa (1984) found a 20 °C higher thallus temperature of the dark-colored lichen

species *Bryocaulon divergens* than the light-colored lichen species *Alectoria ochroleuca*. Parallel to our findings, Gersony et al. (2016) measured with infrared thermography that the leaf temperature of *B. nana* is lower than for any other species in plots from a range of tundra types located in northern Alaska. A possible explanation could be that a difference in canopy morphology between lichens and shrubs leads to differences in the energy balance (i.e. evapotranspiration, see Sect. 4.2.3, Gersony et al., 2016), but this cannot be concluded from our measurements. Therefore, more research including the full energy

balance and surface temperature of vegetation is needed to draw a solid conclusion on the dynamics between the albedo and the surface temperature of lichens and other vegetation types.

**4.2 Subsurface microclimate**

The higher soil heat flux and soil temperature underneath the lichen plots during nearly the entire field season was not in line with our hypothesis. We thus infer that the higher albedo is not generating a cooler subsurface compared to shrubs, but that

other differences between lichens and shrubs are more determinative. Our results are supported by the study of Mikola et al.





(2018), who also found lower soil temperatures below shrub plots than below lichen plots in the Siberian arctic tundra. We consider three reasons that might lead to a lower soil heat flux and soil temperature below shrubs compared to lichens: (1) differences in the subsurface between lichens and shrubs (i.e. litter layer); (2) shading of the surface underneath the taller shrubs with dense canopy; (3) differences in the energy balance of lichens and shrubs. In the following sections we will discuss

how these three reasons might have affected our measurements.

### 4.2.1 Litter layer

A difference in subsurface between the lichen and shrub plots could lead to differences in the thermal properties (e.g. thermal conductivity, heat capacity) of the subsurface (Abu-Hamdeh and Reeder, 2000; Oke, 2002), which can in turn affect the soil heat flux and soil temperature. Since we did not find a difference in several mineral soil properties nor a difference in soil

moisture between the lichen and the shrub plots (Fig. 3), it is unlikely that differences in the mineral soil initiated the higher soil heat flux and soil temperature below lichens. However, we did find a large difference in litter layer thickness below lichens and shrubs. This organic litter layer contains among others dead leaves and roots and has a low thermal conductivity (Abu-Hamdeh and Reeder, 2000). Due to this low conductivity, the litter layer has an insulating effect on the mineral soil underneath and is likely to lead to a lower soil heat flux and soil temperature below shrubs (Beringer et al., 2001). Only some studies have

addressed the insulating capacity of litter in the field (Beringer et al., 2001; Barrere et al., 2017). For example, Barrere et al. (2017) measured a thermal conductivity of 1.36 W m$^{-1}$ K$^{-1}$ for an arctic soil and 0.19 W m$^{-1}$ K$^{-1}$ for a dry litter layer from shrubs in the Canadian Arctic and simulated that this litter layer decreased the summer soil temperature considerably. The insulating properties of litter might be of specific interest for our study, since the thermal conductivity is mainly depending on moisture availability (De Vries, 1963; Ochsner et al., 2001; Oke, 2002). Since our measurements were conducted during a relatively

dry summer, the thermal conductivity of the litter might be even lower than during a normal summer and therefore the insulating effect might have been amplified.

The insulating properties of litter can potentially also explain the smaller daily amplitude and delay in maximum and minimum for the soil heat flux and soil temperature below shrubs (Fig. 8). It appears that the soil below lichens gains and loses heat much easier than the soil below shrubs and that the soil temperature below lichens is more strongly coupled with the air

temperature than the soil temperature below shrubs. Figure 5c also indicates a stronger relationship between soil- and air temperature at the lichen plots than at the shrubs plots.

### 4.2.2 Shading by the shrub canopy

A second reason that we consider likely to cause the lower soil heat flux and soil temperature below shrubs compared to lichens, is the shading effect of the subsurface by the shrub canopy. Loranty et al. (2018) state that the amount of energy

available for the soil heat flux depends among others on the thermal gradient between the ground surface and the underlying soil. They advocate that the temperature of the ground surface, which includes only ground-cover vegetation such as lichens and mosses, is a better variable than the temperature of the land surface, which includes tall overlying vegetation canopies,





since it is the ground-cover vegetation that is in direct contact with the underlying soil. It is plausible that the ground surface temperature in our lichen plots was higher than the ground surface temperature in our shrub plots due to shading of *Betula nana* on the ground cover vegetation in the shrub plots. Therefore the thermal gradient between the ground cover vegetation and the soil was likely to be larger in our lichen plots and this led subsequently to a higher soil heat flux.

Multiple studies have addressed the shading effect of shrub canopies (Bewley et al., 2007; Juszak et al., 2014; Williams et al., 2014; Juszak et al., 2016). Juszak et al. (2016) measured an average growing season transmittance of only 0.36 below *Betula nana*. Moreover, multiple studies have also measured the impact of this shading on the soil microclimate below shrubs (Blok et al., 2010; Blok et al., 2011; Myers-Smith and Hik, 2013; Juszak et al., 2016). Blok et al. (2010) measured a lower $Q_G$ below *Betula nana* plots with a dense canopy compared to plots where the canopy was removed, despite a higher net radiation for the plots with a dense canopy. This low $Q_G$ led to a decrease in active layer thickness below the plots with a dense canopy. Also, Myers-Smith and Hik (2013) found a 2 °C lower soil temperature below shrub plots compared to tundra plots due to shading of the shrub canopy. The above-mentioned studies show the possibility that shading by shrubs can cause a lower soil heat flux and soil temperature and this might therefore have led to a higher soil heat flux and soil temperature below our lichen plots compared to our shrub plots.

### 4.2.3 Energy balance

A third reason for the unexpected results that we consider, is that a larger part of the net radiation of the shrub plots is used for evapotranspiration compared to the lichen plots and therefore a smaller fraction of net radiation is left to heat the soil. The net radiation that is available at the earth surface is usually partitioned over three components (Eq. 4) (Oke, 2002):

$$Q^* = Q_H + Q_E + Q_G \tag{4}$$

in which $Q_H$ is the energy that is used to heat up the atmosphere (sensible heat flux), $Q_E$ is the energy that is used for evapotranspiration (latent heat flux), and $Q_G$ is the energy that penetrates into the soil (soil heat flux). Since lichens do not have roots, they will not take up water actively from the soil and transpire back in the atmosphere. Therefore, the latent heat flux over a lichen canopy is solely dependent on evaporation and is relatively low. This is in contrast to shrub vegetation that does take up and transpire water actively. The latent heat flux of shrubs is the sum of evaporation and transpiration and can thus be expected to be relatively high.

To verify that a larger part of the available energy is used for $Q_E$ of our shrub plots compared to our lichen plots, we estimated $Q_E$ of the plots measured in 2019 in a similar way as Eaton et al. (2001), using the formula of Priestley and Taylor (1972). Table 3 shows how the net radiation is partitioned over the three fluxes in our plots (see Appendix F for calculations and additional results). These values imply that shrubs do use a larger part of the net radiation for evapotranspiration than lichens. The fraction $Q_E/Q^*$ of 0.55 for our lichen plots is close to the 0.49 found by both Eaton et al. (2001) and Boudreau and Rouse (1995) for a lichen-heath tundra. Moreover, the Bowen ratios fall within the range of low arctic upland tundra and low arctic shrub tundra reported by Eugster et al. (2000). Multiple other studies have found a higher $Q_E/Q^*$ for shrub tundra compared to shrub-free tundra heaths (McFadden et al., 1998; Eugster et al., 2000), even though their tundra heaths consisted





**Table 3: The average partitioning of net radiation over the three fluxes and the average Bowen ratio ($Q_H/Q_E$) for the three paired lichen and shrub plots of 2019. See Appendix F for the calculation and additional results.**

|  | Lichen | Shrub |
|---|---|---|
| $Q_E/Q^*$ (-) | 0.55 | 0.68 |
| $Q_H/Q^*$ (-) | 0.36 | 0.27 |
| $Q_G/Q^*$ (-) | 0.09 | 0.05 |
| Bowen (-) | 0.68 | 0.42 |

of less lichens than our study. Moreover, Sabater et al. (2020) measured that lichens have less evapotranspiration than shrubs,

when both vegetation types are part of the understory vegetation of a mountain birch forest. Therefore, we infer that the relatively high latent heat flux for our shrub plots is also a potential reason for their lower soil heat flux and soil temperature compared to our lichen plots. In addition, the lower $Q_H/Q^*$ for the shrub plots could be an explanation for the lower $LW_{out}$ and inferred lower surface temperature of the shrub plots compared to the lichen plots (see Sect. 4.1). However, measurements on the complete energy balance of lichens and shrubs are needed to confirm the partitioning of the net radiation over the three

heat fluxes.

**4.3 Synthesis**

It is not possible from our study to conclude if one or more of the proposed reasons lead to the lower soil heat flux and soil temperature below shrubs compared to lichens. Considering the delay in maximum and minimum soil heat flux and the lower daily amplitude of the soil heat flux and soil temperature below shrubs (Fig. 8), it is likely that the litter layer below the shrubs

plays an important role, since the other two proposed reasons would not generate this effect. However, additional measurements are needed to give a solid conclusion. Nevertheless, our study does show that the high albedo of lichens is not leading to lower soil temperatures below lichens than below shrubs during the growing season. Since recent studies have shown that differences in color, and therefore albedo, did not even cause a difference in soil temperature between lichen species (Nystuen et al., 2019; Van Zuijlen et al., 2020), it is unlikely that albedo is an important factor determining the difference in

soil temperature between the two different vegetation types. Instead, the marked differences in canopy structure between lichens and shrubs are a more essential factor.

Our study shows that a shift from lichens to shrubs decreases the summer soil temperature, while other studies showed that shrub expansion can lead to higher winter soil temperatures, since the shrub canopy is trapping snow that insulates the soil (Sturm et al., 2001a; Myers-Smith and Hik, 2013). As a result, a shift from lichen heaths to shrub vegetation leads to lower

soil temperature fluctuations during the course of a year. The change in fluctuation will be even more distinct with a shift from lichen heaths to shrub vegetation than with a shift from a general arctic tundra towards shrub vegetation. Reason for this is that lichen heaths occur mainly on areas with shallow or missing snow cover, which are characterized by low winter soil temperatures (Odland and Munkejord, 2008; Sundstøl and Odland, 2017). In addition, a shift towards shrubs might have



important consequences for permafrost, soil microbial activity and carbon storage due to a changing soil temperature (Myers-
Smith et al., 2011; Loranty et al., 2018). Therefore, these consequence might be more distinct when the initial vegetation stage
includes a large abundance of lichens, since the soil temperature change is larger.

Multiple studies have discussed the cooling capacity of lichens on the underlying soil, and have argued that this
cooling capacity is a result of their high albedo and the insulating properties of lichens due to their low conductivity. However,
most studies measured or modelled lower soil temperatures below lichens in relation to bare soil or disturbed lichens (Beringer
et al., 2001; Gold et al., 2001; Porada et al., 2016; Nystuen et al., 2019; Van Zuijlen et al., 2020), but the comparison with
another vegetation type has rarely been made. Our study shows that the cooling capacity of lichens does not lead to a lower
soil temperature compared to shrubs. In addition, Van Zuijlen et al. (2020) concluded that the difference in soil microclimate
between lichen species is not driven by the color of lichen species, but by lichen mat morphology. Therefore, our study and
the study of Van Zuijlen et al. (2020) imply that the insulating capacity of lichens is a much more important factor determining
the soil temperature than the high albedo.

Although the high albedo of lichens does not have a cooling effect on the subsurface, it will have a cooling effect on
the atmosphere. Since lichens might continue to decrease in abundance due to shrub expansion, it is important to estimate the
impact of such a shift on regional and possibly global climate. We measured an average increase in atmospheric heating of
3.35 MJ per day during the growing season with every square meter of lichen that turns into shrub. This value is among others
dependent on the incoming solar radiation and can therefore change with latitude and day of the year. Modelling studies should
use our measurements to estimate the impact of the loss of lichen cover on the climate over alpine and arctic areas.

## 5. Conclusion

Our study shows that lichens have a lower net radiation than shrubs during the growing season. In addition, we show that the
soil underneath the lichens has a higher soil temperature and a higher soil heat flux than the soil below shrubs, especially
during warm days. This implies that the relatively high albedo of lichens affects the radiation balance, but not the subsurface
microclimate. Potential reasons for this could be the thicker litter layer, shading by the canopy or more evapotranspiration in
the shrub plots. We conclude that the decline of lichens due to shrub expansion will lead to atmospheric heating (i.e. higher
latent + sensible heat flux), but has a cooling effect on the subsurface during the growing season. Future studies should focus
on the quantification of the effect of lichen decline on the climate on a regional and possibly on the scale of the arctic.





**Appendix A**



**Figure A1: Picture of the study area. Lichens heaths are mainly located on the windswept ridgetops, while shrubs are located on the midslopes and ridgetops.**





## Appendix B

**Table B1: Sensors of the HOBO RX 3000 remote weather station that was used during the field season of 2019.**

| Variable | Sensor | Height | Measurement interval |
|---|---|---|---|
| Air temperature | 12-bit Temperature/Relative humidity smart sensor (S-THB-M002) | 0.5, 1, 2 m | 5 min |
| Relative humidity | 12-bit Temperature/Relative humidity smart sensor (S-THB-M002) | 0.5, 1, 2 m | 5 min |
| Precipitation | 0.2 mm Rainfall smart sensor (S-RGB-M002) | 3 m | Total over 5 min |
| Incoming solar radiation | Solar radiation (silicon pyranometer) smart sensor (S-LIA-M003) | 3 m | 5 min |
| Wind speed | Wind speed smart sensor (S-WSB-M003) | 3 m | Average over 5 min |
| Wind direction | Wind direction smart sensor (S-WDA-M003) | 3 m | Average over 5 min |
| Barometric pressure | Smart barometric pressure sensor (S-BPB-CM50) | - | 5 min |
| Data logger | HOBO RX3000 remote monitoring station data logger (RX3000) | - | - |



**Appendix C**

**Figure C1: Daily measurements during the field season of 2019 for the background weather conditions (air temperature and precipitation) (a), net radiation (b), soil heat flux (c), soil temperature (d), and soil moisture (e). Every day is divided into the lichen and corresponding shrub plot for graphs (b), (c), (d), and (e). For ease of display, we calculated averages per plot for the soil heat flux (n=2), soil temperature (n=3) and soil moisture (n=3). Error bars display minimum and maximum values for soil heat flux and standard errors for soil temperature and soil moisture.**




**Appendix D**

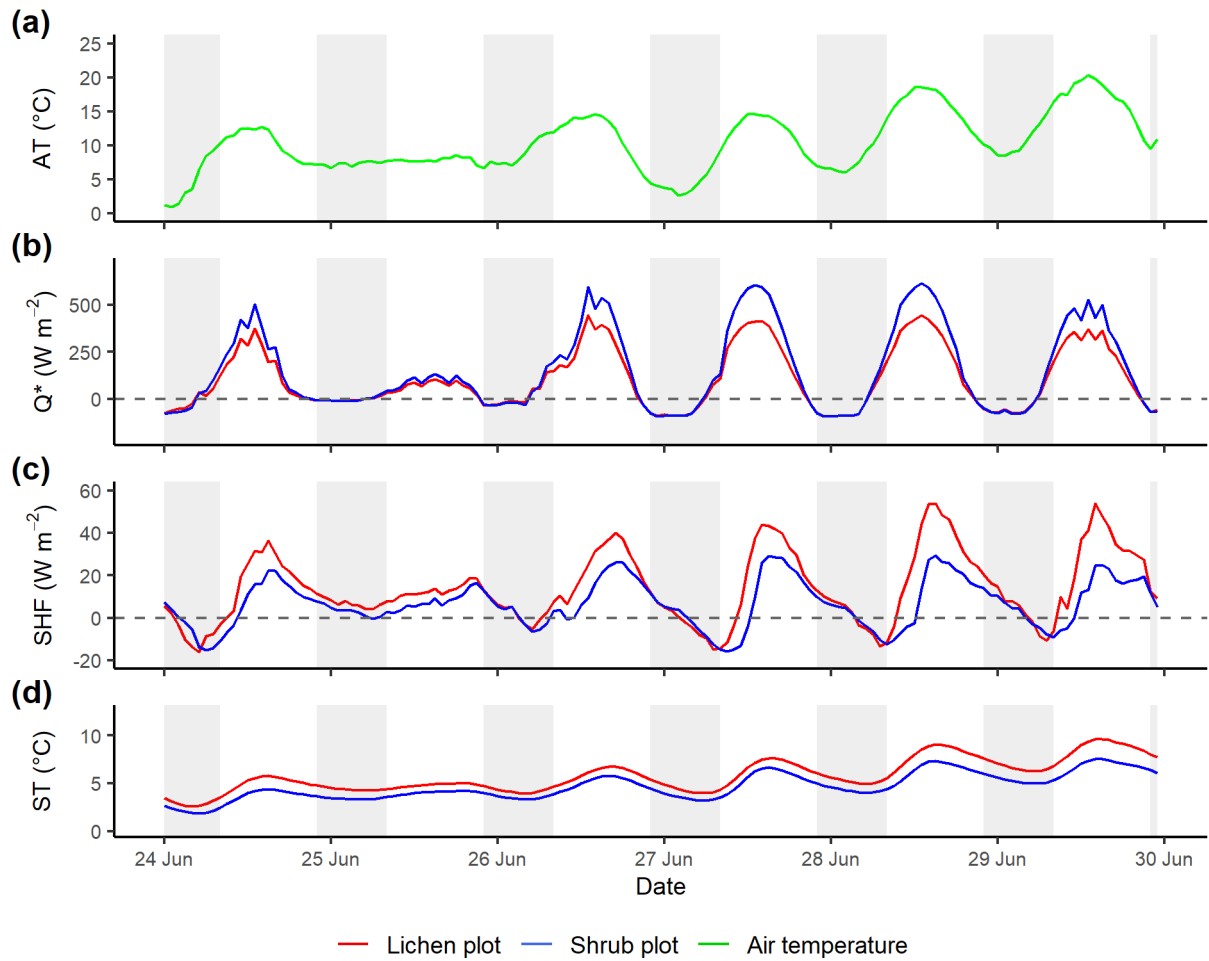

**Figure D1: Time series of the reference air temperature (a), net radiation (b), soil heat flux (c) and soil temperature (d) of paired plots 1 of 2019. Time series are constructed with hourly averages. Shaded areas indicate nighttime (22:00– 08:00 LT). AT = reference air temperature, Q\* = net radiation, SHF = soil heat flux, ST = soil temperature.**



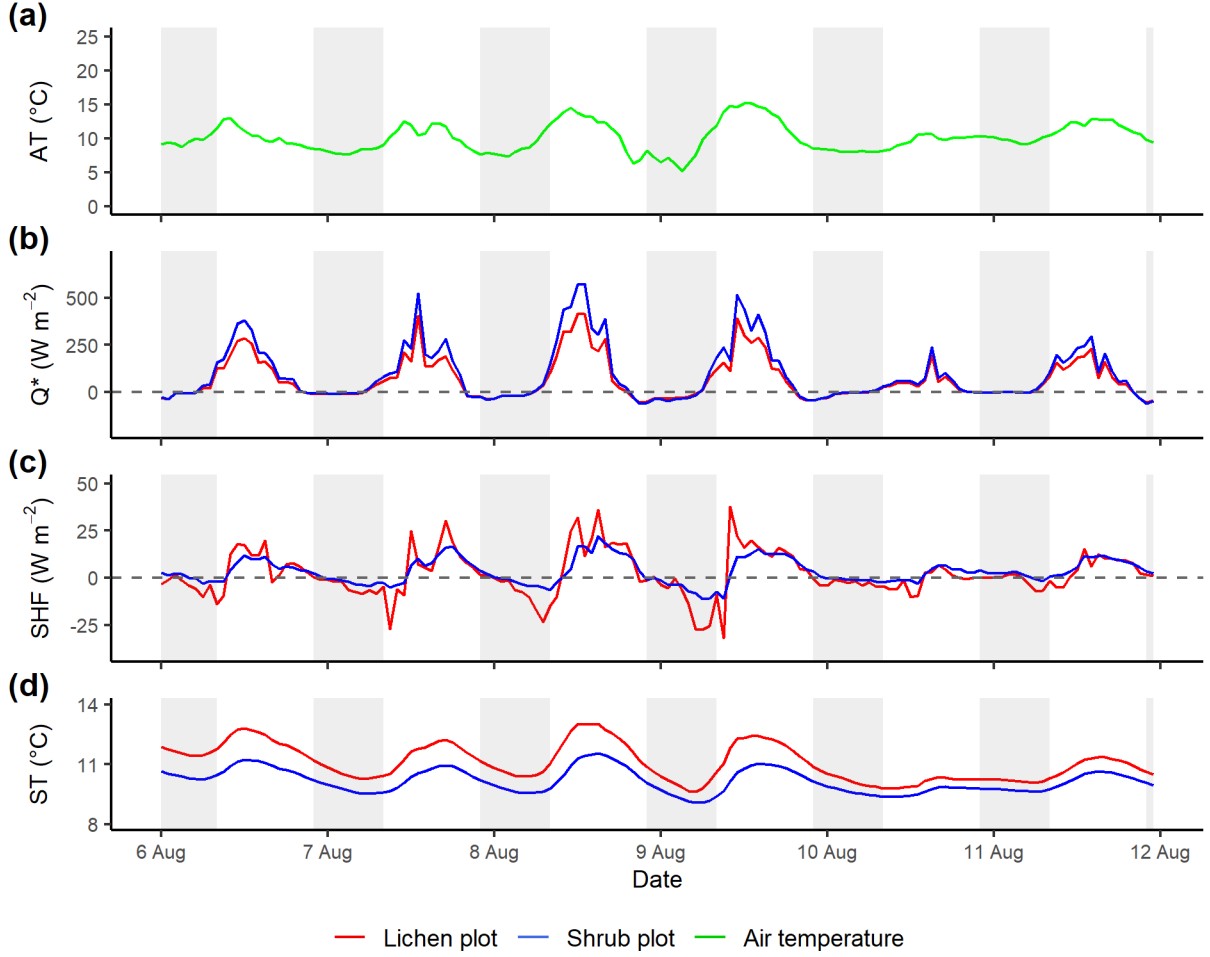

**Figure D2:** Time series of the reference air temperature (a), net radiation (b), soil heat flux (c) and soil temperature (d) between paired plots 3 of 2019. Time series are constructed with hourly averages. Shaded areas indicate nighttime (22:00–08:00 LT). AT = reference air temperature, Q* = net radiation, SHF = soil heat flux, ST = soil temperature.





**Appendix E**

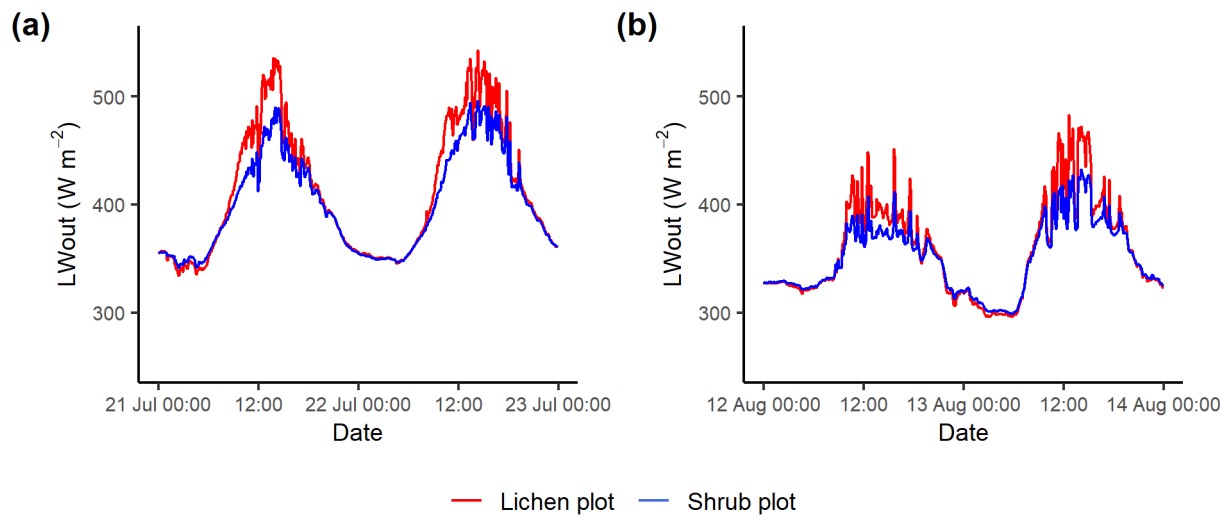

**Figure E1: Time series of the outgoing longwave radiation of the paired lichen and shrub plots measured on 21 and 22 July 2018 (a) and on 12 and 13 August 2018 (b).**





**Appendix F**

To test if the shrub plots used more energy for the latent heat flux than the lichen plots, we calculated the latent heat flux for

the plots of the field season of 2019 in a similar way as Eaton et al. (2001) using the formula of Priestley and Taylor (1972):

$$Q_E = \alpha \times (S/(S + \gamma)) \times (Q^* - Q_G) \qquad (1)$$

in which $\alpha$ is an empirical constant (-), S is the slope of the saturation vapour-temperature curve (Pa K$^{-1}$) depending on the air

temperature and $\gamma$ is the psychrometric constant (65 Pa K$^{-1}$). We used the $\alpha$ for upland lichen-heath tundra (0.90) and shrub

tundra (1.08) estimated in the Canadian Arctic for our estimation (Eaton et al. 2001). Subsequently, we calculated the sensible

heat flux with:

$$Q_H = Q^* - Q_E - Q_G \qquad (2)$$

and the Bowen ratio, with:

$$\beta = Q_H/Q_E \qquad (3)$$

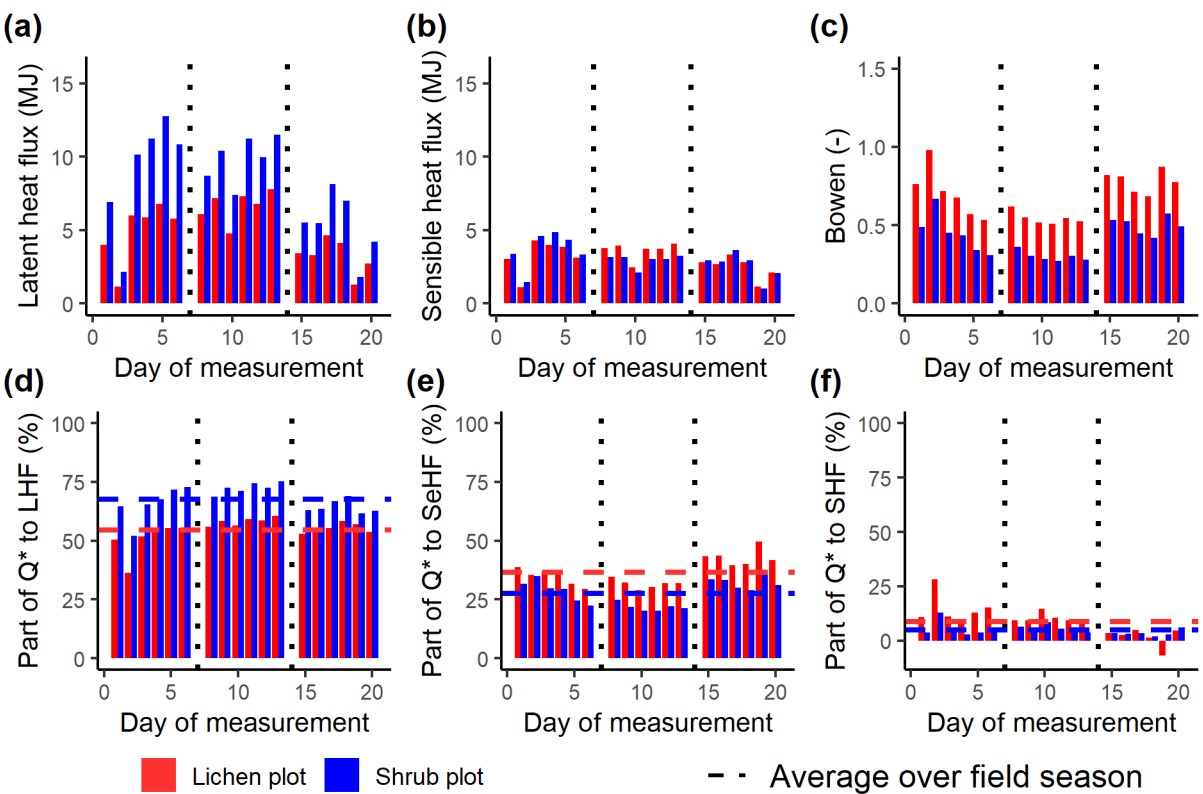

**Figure F1: The calculated latent heat flux (a), sensible heat flux (b), Bowen ratio (c) and the fractions $Q_E/Q^*$ (d), $Q_H/Q^*$ (e) and $Q_G/Q^*$ (f) per lichen and shrub plot of 2019. The vertical dotted lines indicate a plot change. LHF = latent heat flux, SeHF = sensible heat flux, SHF = soil heat flux, $Q^*$ = net radiation. See Appendix C for the exact dates of the measurements of the three paired plots.**





## Data availability

The data will be made available on the open-access research repository (Figshare) of the University of South-Eastern Norway after acceptance.

## Author contribution

PA, AO, SR and HR conceived the study design. PA collected the data. PA analyzed the data with input from JA. PA wrote the manuscript with contributions from all other authors.

## Competing interests

The authors declare that they have no conflict of interest

## Acknowledgements

We thank Roland Pape for his valuable comments on an earlier version of this manuscript. This study was funded by the University of South-Eastern Norway. This study was also supported by a grant from the Research Council of Norway (249902/F20) and Nansenfondet to JA. We thank the Statsminister Gunnar Knudsen og hustru Sofie født Cappelens familielegat for the funding of the weather station.





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
