# Peer review of "Microclimatic comparison of lichen heaths and shrubs: shrubification generates atmospheric heating but subsurface cooling during the growing season"

_Biogeosciences, 2020_

## Referee Comment (RC1) · Anonymous Referee #1 · 25 Nov 2020

This paper investigates the effect of lichen cover on alpine tundra microclimates, relative to increased shrub cover growth. The authors use two years worth of vegetation and microclimate data collected in Alpine Norway and find that a decline in lichens and increase in shrub cover lead to an overall atmospheric heating effect. I think this is a really well framed research gap highlighting an understudied and potentially very important research gap. While the data only come from one site, I think the topic is in the scope of Biogeosciences, as it has potential tundra-wide implications. I think the paper is overall clearly written, with a thorough methodology, and interesting discussion. However, I propose that this paper be reconsidered after major revisions because I feel like more work could be done to frame the research in the context of ecosystem-wide

change, and I think more work could be done to frame hypotheses from the beginning of the paper.

The introduction and literature review is concise and persuasive, but I feel as though you could state clear hypotheses at the end of this section to better frame the rest of the paper. You have nice comprehensive microclimate measurements, and your methods section is replicable and thorough. I think your results section works well although I have provided some comments on the figures. Finally, I'd like to see more discussion of the implications of these findings for other alpine and Arctic tundra sites - do you expect to see similar trends across the biome? What significance do you think this has for the ecosystem as a whole? Do you expect to see similar trends in the Arctic tundra too, or just Alpine sites - and what is your justification for this? Finally, how much do you think shrub species matter in explaining the variation in your data? Are all tundra shrubs out-competing all lichens? I think this could be an interesting line of enquiry.

I'm looking forward to reading this paper again!

LINE COMMENTS: - generally, you should capitalise 'Arctic' -21: "we hypothesise" - reword. The placement of this phrase in the abstract makes it seem like a hypothesis you set at the beginning relating to this project as opposed to a theory you generated based on your results - 33: "this region" - I understand from context, but you could make it clearer which region you mean in this sentence (i.e. are you still talking about the Western Canadian Arctic, or Alpine Norway again?) - 36: not clear from this sentence if lichen is declining purely because of competition with shrubs, or if they aren't coping physiologically with a rapidly warming temperature - 61: I'd call it "macroclimate" instead of "large climate" - Figure 2:: could you add the letter labels to the photography too? Its currently quite hard to envision. - 161: be explicit here, what is the interaction and why did you include it? We get the interaction later on in the tables but I feel you could spell it out here. - Tab 2: maybe you should also add a column saying how much variation in % is explained by your random effects? Also you should add the slope/estimate and error in this table, as the F and P values are not very informative

on their own. - Figure 4: a nice clear figure. Move the (d) label up a bit, it looks a bit cluttered at the moment. - Figure 5: looks great but I'd recommend changing the colours from red to blue to something less indicitive of temperature. Currently it suggests lichens = hot, shrubs = cold. - Figure 8: useful site information, but maybe move to supplementary materials? You have a lot of figures and I feel like this one can be moved. - 298: "tundra plots" vs "shrub plots" = this is confusing because tundra can be very shrubby. Maybe rename these descriptors to make this more explicit?

---

## Referee Comment (RC2) · Anonymous Referee #2 · 30 Nov 2020

This paper investigates the differences between lichen and shrub cover for alpine tundra microclimates. The authors measured vegetation and microclimate data at a site in Norway in two consecutive years. The research gap that they are aiming to investigate is well-framed and interesting for the field of tundra ecology in a wider scope. The article is well-written, especially the discussion and methods are overall clear. The main strength of the paper in my view is that it introduces meticulous measurements on an understudied aspect of this field. However, I propose major revisions to this paper before acceptance. The authors should reframe the conclusions of their study in accordance to the size and caveats of their experiment and improve on their analysis of the results.

[Figure]

The paired plot set-up, and the plot selection not being random, risks the introduction of a selection bias. Despite their thorough methodology it remains unclear as to how their plots where actually selected. Was selection bias prevented through radial/cardinal walking? How was a plot determined to be on a ridgetop, and how was a slope <10% determined? Why was this area picked in general, does it represent the landscape that they want to study in a particularly good way? The authors successfully performed very high-detail measurements for both radiation and soil parameters. The measurements where however always on different days for different plots, and there was no revisiting of same plots, not even in 2019 when they revisited the study site. This limited amount of measuring days, measured plots and no revisiting of plots increases the risk of the results being affected by unaccounted variability. 2018 for instance was a very warm and dry year that could influence many processes in the ecosystem. Please expand on the potential consequences of these limitations and why the experiment design was changed in 2019. The authors perform their statistics based on a sample size of 13, Wilcoxon is however not a parametrized method, and generally not suitable for small sample sizes. Significance of results based on these tests can't be concluded with so much certainty as the authors claim. Consider trying a paired parametric test. Be aware of confounding variables in the microclimate.

Comments on figures: the amount of figures in the paper could be decreased. Fig. 3 needs better axis scaling to highlight their results. Fig. 5 could benefit from an r2 to indicate the strength of relationship. In Fig. 7 it is hard to read the confidence intervals due to overlapping the same colors. There is serious cherry-picking in the 2019 graphs in fig. 8 and fig. 9. In 2019 only three plots were measured and then the authors select the one that support their conclusions the best to include in the paper. The supporting figures of the remaining 2 plots that were measured in 2019 don't show the same strong results as the ones that are included in the main paper.

Generally, it is a good study with only 13 plots over two years, and only a few consecutive days of measurements. 2018 was a very warm and dry year. Their plot selection

could have introduced bias. The authors should make their claims according to the limits of their experiment. The same group of authors have published multiple papers focusing on slightly different aspects of this same site and experiment, further decreasing the novelty of the results presented here. My suggestion would be to extend the study by revisiting the sites in a different year, or performing the same experiment at a different site. The results would be very much strengthened by such an extension of the study and the scientific community would benefit from a more thorough investigation of this interesting experiment.

General comments: l.32 add a Norwegian or at least Scandinavian example? l.35 what about increased competition due to e.g. grazing? l.42 does vegetation composition indicate the vegetation community, abundance, diversity? l.47 what other distinct characteristics other than albedo? l.81 expand on insulating properties of lichens? Include differences between e.g. crustose and foliose lichens and indicate which type of lichen species were dominant in their plots? l. 108 'similar way, similar positions', explain how exactly? l.111 30cm above canopy. So much higher for shrubs with a much higher canopy? Consider other effects such a surface roughness and turbulence affecting this. Tbl.2 move to supplementary information

---

## Referee Comment (RC3) · Anonymous Referee #3 · 30 Nov 2020

This study examines differences in surface energy partitioning and soil microclimate between lichen- and shrub-dominated vegetation in southern Norway using paired measurements made with a set of mobile instrument platforms. The authors find that the lower albedo of shrub canopies leads to higher atmospheric heating, but lichen mats have greater soil heat fluxes and temperatures despite lower net radiation. The latter is attributed shrub canopy shading and a thicker litter layer with lower thermal conductivity. The results provide important context for understanding how shrub expansion will affect microclimate when shrubs replace lichens. The paper is well written, interesting, and I enjoyed reading it. There are several improvements that could help to strengthen the paper before it is considered further for publication.

[Figure]

While I don't think there are any technical problems with how the methods were applied, I do find it curious that different approaches were used to analyze the data from different years, and that the data weren't aggregated. Why not use data from 2018 and 2019 in the mixed models to examine differences in microclimate (i.e. in Table 2)? A box and whisker plot showing mean microclimate by vegetation type would be more informative than the individual daily averages shown in Figure 4. Something like Figure 3, but instead showing net radiation, soil heat flux, soil temperature, etc... I also wonder whether it would be more appropriate to model microclimate conditions as a function of incident shortwave radiation, rather than temperature, since this likely affects soil temperature and heat flux more so than air temperature (e.g. L163-165)? Related, are Figure 5 & 7 showing results of the mixed effects models?

Also I recommend that the authors consult Loranty et al 2018 published in PLOS One. This paper examines differences in soil temperature, thermal conductivity, surface temperature, and ET between lichen- and shrub-dominated vegetation patches in Siberia. It reaches many of the same conclusions presented in this manuscript, and would provide useful context in the introduction and discussion.

Loranty, M.M., Berner, L.T., Taber, E.D., Kropp, H., Natali, S.M., Alexander, H.D., Davydov, S.P. and Zimov, N.S., 2018. Understory vegetation mediates permafrost active layer dynamics and carbon dioxide fluxes in open-canopy larch forests of northeastern Siberia. Plos one, 13(3), p.e0194014.

Minor comments: L1: Does your study really address the decline of lichen heaths? The results certainly have implications in this context, but it seems more like a comparison between lichen heaths and shrubs. Something to that effect would be more appropriate in the title.

L45-50: This seems like it refers to another study that uses the same data presented in this manuscript. It would be appropriate to note that.

L80: It should be made clear here that the authors know albedo is higher for lichen as

a result of previous analyses from this data set.

L85: Is there permafrost at this site?

L216: The wording "As for the net radiation" is a little confusing for me.

L229: It is worth noting here that alpine tundra and lichen mats will also have different albedos, and the while lichen does have high albedo and can be somewhat abundant, it is not broadly representative of alpine tundra.

L240: See Loranty et al 2018 in PLOS One for surface temperature and ET measurements of lichens vs shrubs.

L303: ET would affect the canopy temperature more than that ground temperature, and as noted on line 285 the latter likely has more impact on soil temperature. However, higher ET shrubs may cool canopy temperatures, meaning less LW emitted from the canopy, and LW enhancement by canopies can affect the energy balance at the ground surface (e.g. Todt et al, 2018; Wake et al, 2017). This hasn't been shown in shrub tundra, but might be worth considering here.

L361-365: This section could be developed a bit more. It would be worth discussing how pervasive lichens are across alpine and arctic regions more generally. What types of modeling studies might your measurement help to inform, ecosystem or global scale studies, are there any example citations? Which measurements specifically might be useful for modeling?

Figure 7: Why aren't the points included here, but included in Figure 5?

Figure 9: The abbreviations ST, SHF, etc... should be used consistently throughout the figures/manuscript.

References: Todt, M., Rutter, N., Fletcher, C.G., Wake, L.M., Bartlett, P.A., Jonas, T., Kropp, H., Loranty, M.M. and Webster, C., 2018. Simulation of longwave enhancement in boreal and montane forests. Journal of Geophysical Research: Atmospheres,

123(24), pp.13-731.

Webster, C., Rutter, N. and Jonas, T., 2017. Improving representation of canopy temperatures for modeling subcanopy incoming longwave radiation to the snow surface. Journal of Geophysical Research: Atmospheres, 122(17), pp.9154-9172.
* * *

---

## Author Comment (AC1) · 18 Dec 2020

We thank Reviewer 1 for his/her time to review our manuscript and for his/her valuable comments. Below are our answers on his/her comments.

Reviewer comment: This paper investigates the effect of lichen cover on alpine tundra microclimates, relative to increased shrub cover growth. The authors use two years worth of vegetation and microclimate data collected in Alpine Norway and find that a decline in lichens and increase in shrub cover lead to an overall atmospheric heating effect. I think this is a really well framed research gap highlighting an understudied and potentially very important research gap. While the data only come from one site, I think

the topic is in the scope of Biogeosciences, as it has potential tundra-wide implications. I think the paper is overall clearly written, with a thorough methodology, and interesting discussion. However, I propose that this paper be reconsidered after major revisions because I feel like more work could be done to frame the research in the context of ecosystem-wide change, and I think more work could be done to frame hypotheses from the beginning of the paper.

Author response: We thank the Reviewer for his/her positive words concerning the importance of our study and the writing style of the paper. Below we reply in detail to his/her comments concerning the framing of the hypotheses and the framing of our research in the context of ecosystem-wide change.

Reviewer comment: The introduction and literature review is concise and persuasive, but I feel as though you could state clear hypotheses at the end of this section to better frame the rest of the paper. You have nice comprehensive microclimate measurements, and your methods section is replicable and thorough. I think your results section works well although I have provided some comments on the figures. Finally, I'd like to see more discussion of the implications of these findings for other alpine and Arctic tundra sites - do you expect to see similar trends across the biome? What significance do you think this has for the ecosystem as a whole? Do you expect to see similar trends in the Arctic tundra too, or just Alpine sites - and what is your justification for this? Finally, how much do you think shrub species matter in explaining the variation in your data? Are all tundra shrubs out-competing all lichens? I think this could be an interesting line of enquiry. I'm looking forward to reading this paper again!

Author response: We thank the Reviewer for his/her positive feedback concerning the introduction, methods and results section. In the new version of the manuscript, we will list the hypotheses at the end of the introduction in order to frame the rest of the paper. We will also discuss more extensively the more general implications of our findings for other alpine and Arctic areas. We see that this is necessary since our measurements were conducted on one field site. To elaborate on the question if we

expect similar trends across the biome, we will include more studies that have been measuring microclimatic conditions below lichens and shrubs in alpine and Arctic areas before (e.g. Grünberg et al., 2020; Loranty et al., 2018; Mikola et al., 2018). Moreover, we will use articles that studied the implications of shrubification on alpine and Arctic areas (e.g. Chagnon & Boudreau, 2019; Frost et al, 2018; Lafleur & Humphreys 2018), to discuss if we expect different results between different tundra shrubs species and shrub characteristics (e.g. shrub height, shrub density) concerning the variation in microclimate between lichens and shrubs. We agree that this additional discussion is of relevance for our manuscript.

Reviewer comment: - generally, you should capitalise 'Arctic'

Author response: We will capitalize "Arctic" in the new version.

Reviewer comment: -21: "we hypothesise" - reword. The placement of this phrase in the abstract makes it seem like a hypothesis you set at the beginning relating to this project as opposed to a theory you generated based on your results

Author response: We see the point that the Reviewer makes and will replace "hypothesize" with "predict".

Reviewer comment: - 33: "this region" - I understand from context, but you could make it clearer which region you mean in this sentence (i.e. are you still talking about the Western Canadian Arctic, or Alpine Norway again?)

Author response: We will make this clearer.

Reviewer comment: - 36: not clear from this sentence if lichen is declining purely because of competition with shrubs, or if they aren't coping physiologically with a rapidly warming temperature

Author response: We will elaborate more on this in the new version.

Reviewer comment: - 61: I'd call it "macroclimate" instead of "large climate"

Author response: We will replace "large-scale climate" with "macroclimate".

Reviewer comment: - Figure 2:: could you add the letter labels to the photography too? Its currently quite hard to envision.

Author response: We will add the letter labels in the photography of Figure 2.

Reviewer comment: - 161: be explicit here, what is the interaction and why did you include it? We get the interaction later on in the tables but I feel you could spell it out here.

Author response: We will elaborate more on the interaction term in the new version of the manuscript and explain why we included it.

Reviewer comment: - Tab 2: maybe you should also add a column saying how much variation in % is explained by your random effects? Also you should add the slope/estimate and error in this table, as the F and P values are not very informative on their own.

Author response: We will add the slope and error in Table 2 and will add a column saying the marginal and conditional r-squared for each model. The marginal r-squared is the variation explained by the fixed effects and the conditional r-squared is the variation explained by the entire model. Since Table 2 will get rather large when we add this information for the daily, daytime and nighttime analysis, we will report only the table with the analysis of the daily averages/totals in the main text and will place the table with the analysis of the daytime and nighttime averages/totals in the Appendix.

Reviewer comment: - Figure 4: a nice clear figure. Move the (d) label up a bit, it looks a bit cluttered at the moment.

Author response: We will move the (d) label a bit up.

Reviewer comment: - Figure 5: looks great but I'd recommend changing the colours from red to blue to something less indicitive of temperature. Currently it suggests

lichens = hot, shrubs = cold.

Author response: We will change the colors in all the figures from red and blue into light-green (for lichens) and dark-green (for shrubs). See Figure 1 below for an example.

Reviewer comment: - Figure 8: useful site information, but maybe move to supplementary materials? You have a lot of figures and I feel like this one can be moved.

Author response: We will move Figure 8 to the Appendix.

Reviewer comment: - 298: "tundra plots" vs "shrub plots" = this is confusing because tundra can be very shrubby. Maybe rename these descriptors to make this more explicit?

Author response: We see the point made by the Reviewer and will change the term "tundra plots" into "open tundra plots". The plots in line 298 refers to the study of Myers-Smith and Hik (2013), who refer to them as "open plots" in their paper.

References used by author:

Chagnon, C., & Boudreau, S. (2019). Shrub canopy induces a decline in lichen abundance and diversity in Nunavik (Québec, Canada). Arctic, Antarctic, and Alpine Research, 51(1), 521-532.

Frost, G. V., Epstein, H. E., Walker, D. A., Matyshak, G., & Ermokhina, K. (2018). Seasonal and long-term changes to active-layer temperatures after tall shrubland expansion and succession in Arctic tundra. Ecosystems, 21(3), 507-520.

Grünberg, I., Wilcox, E. J., Zwieback, S., Marsh, P., & Boike, J. (2020). Linking tundra vegetation, snow, soil temperature, and permafrost. Biogeosciences, 17(16), 4261-4279.

Lafleur, P. M., & Humphreys, E. R. (2018). Tundra shrub effects on growing season energy and carbon dioxide exchange. Environmental Research Letters, 13(5), 055001.

[Figure]

Loranty, M.M., Berner, L.T., Taber, E.D., Kropp, H., Natali, S.M., Alexander, H.D., Davydov, S.P. and Zimov, N.S., 2018. Understory vegetation mediates permafrost active layer dynamics and carbon dioxide fluxes in open-canopy larch forests of northeastern Siberia. Plos one, 13(3), p.e0194014.

Mikola, J., Virtanen, T., Linkosalmi, M., Vähä, E., Nyman, J., Postagonova, O., Räsänen, A., Kotze, J., Laurila, T., & Juutinen, S. (2018). Spatial variation and linkages of soil and vegetation in the Siberian Arctic tundra–coupling field observations with remote sensing data. Biogeosciences, 15(9).

——————————————————————

[Figure]

**(a)**

**(b)**

**(c)**

**(d)**

— Lichen plot  — Shrub plot

**Fig. 1.** Example of Figure 5 with the new colors.

---

## Author Comment (AC2) · 18 Dec 2020

We thank Reviewer 2 for his/her time to review our manuscript and for his/her valuable comments. Below are our answers on his/her comments.

Reviewer comment: This paper investigates the differences between lichen and shrub cover for alpine tundra microclimates. The authors measured vegetation and microclimate data at a site in Norway in two consecutive years. The research gap that they are aiming to investigate is well-framed and interesting for the field of tundra ecology in a wider scope. The article is well-written, especially the discussion and methods are overall clear. The main strength of the paper in my view is that it introduces meticulous

measurements on an understudied aspect of this field. However, I propose major revisions to this paper before acceptance. The authors should reframe the conclusions of their study in accordance to the size and caveats of their experiment and improve on their analysis of the results.

Author response: We thank the Reviewer for his/her positive words about the framing of the research gap and the writing style of the manuscript. We understand his/her point concerning the size and caveats of our experiment and will elaborate on this in the next version of the manuscript. We will in particular frame our experiment, which is based on one study area, in the context of ecosystem-wide change over the Northern Hemisphere. Below we will expand on our additional thoughts about the size of our experiment and the analysis of the results by addressing the more specific comments.

Reviewer comment: The paired plot set-up, and the plot selection not being random, risks the introduction of a selection bias. Despite their thorough methodology it remains unclear as to how their plots where actually selected. Was selection bias prevented through radial/cardinal walking?

Author response: For the plots of 2018, we selected randomly 10 locations in the study area using ArcMap. Subsequently, we selected the lichen plots within a radius of 50 meters around each random location that fitted the criteria in Table 1. In case we found multiple lichen plots that fitted these criteria, we selected the lichen plots with the highest percentage of lichens. Subsequently, we selected the shrub plots within a radius of 50 meters around each selected lichen plot that fitted the criteria in Table 1. In case we found multiple shrub plots that fitted these criteria, we selected the shrub plot with the highest percentage of shrubs. The location of the plots of 2019 were selected subjectively, but still fulfilled the criteria of Table 1. Since the location of the plots of 2019 were selected subjectively, we decided to include only the measurements of 2018 in the mixed effects models and in the Wilcoxon test and therefore bias was prevented in these analyses. We agree that we did not specify the plot selection adequately in the text and will explain this more detailed in the next version of the manuscript.

[Figure]

Reviewer comment: How was a plot determined to be on a ridgetop, and how was a slope <10% determined?

Author response: The ridgetops in our study area are well distinct from the midslopes and the snowbeds (see Appendix A of the manuscript for a picture of the study site). The ridgetops were determined based on their vegetation characteristics and the low or absent slope angle. We obtained the slope of each plot with a clinometer. We will specify this in the methods section of the new version of the manuscript.

Reviewer comment: Why was this area picked in general, does it represent the landscape that they want to study in a particularly good way?

Author response: We selected this area since it has a high abundance of lichens and the landscape is representative for an alpine landscape in Scandinavia. We will elaborate on this in the section "Study area" in the new version of the manuscript.

Reviewer comment: The authors successfully performed very high-detail measurements for both radiation and soil parameters. The measurements where however always on different days for different plots, and there was no revisiting of same plots, not even in 2019 when they revisited the study site. This limited amount of measuring days, measured plots and no revisiting of plots increases the risk of the results being affected by unaccounted variability. 2018 for instance was a very warm and dry year that could influence many processes in the ecosystem. Please expand on the potential consequences of these limitations and why the experiment design was changed in 2019.

Author response: The initial sample design for this study contained only measurement of one field season (2018). However, 2018 turned out to be a warm and dry year and therefore we chose to include measurements of 2019 in this manuscript to show that the measurements of 2019 show a similar pattern (lichen plots having a higher soil temperature and soil heat flux than shrub plots during warm days, despite having a lower net radiation). Unfortunately, we could not revisit the same plots of the field

season of 2018, since the vegetation was disturbed after taking soil samples, and this affects the measurements (especially radiation) substantially. The reason for a change in experimental design was that we wanted to measure the plots for a longer period to gain insights in the variability in the microclimatic conditions over time. In the next version of this manuscript, we will elaborate more on the consequences of the warm and dry field season for our study in the discussion. Moreover, we will give our reasons for a change in experiment design (in Lines 132-135).

Reviewer comment: The authors perform their statistics based on a sample size of 13, Wilcoxon is however not a parametrized method, and generally not suitable for small sample sizes. Significance of results based on these tests can't be concluded with so much certainty as the authors claim. Consider trying a paired parametric test. Be aware of confounding variables in the microclimate.

Author response: As suggested by the reviewer, we repeated the analysis with the paired parametric t-test and the paired permutation test. Both tests draw the same conclusion as the Wilcoxon test. However, in our view, the Wilcoxon test is the appropriate test to use here. The parametric t-test has the underlying assumption that the data has a normal distribution. Since we cannot test this assumption due to the low sample size, we decided to use the more conservative Wilcoxon test. Therefore, we still opt to report the results of the Wilcoxon test. Note: we analyzed the difference in the canopy and soil variables between the lichen and shrub plots (Fig. 3) for the plots of the field season of 2018 only, and therefore the sample size consists of 10 plot pairs. We will make this clear in the new version of the manuscript by adding "n = 10 plots per boxplot" in the caption of Figure 3.

Reviewer comment: Comments on figures: the amount of figures in the paper could be decreased.

Author response: We will move Figure 8 to the Appendix in the next version.

Reviewer comment: Fig. 3 needs better axis scaling to highlight their results.

Author response: We will change the scaling of the y-axis of this figure in the next version.

Reviewer comment: Fig. 5 could benefit from an r2 to indicate the strength of relationship.

Author response: We will determine the marginal and conditional r2 of each linear mixed-effects model as proposed by Nakagawa & Schielzeth (2013) and indicate these r-squares in Table 2. The marginal r-squared is the variation explained by the fixed effects and the conditional r-squared is the variation explained by the entire model. We chose to report these values in Table 2, since Reviewer 1 requested this information in this table. Moreover, we decided to remove the actual measurements from Figure 5 to be consistent with Figure 7, which makes the indication of the r2 in Figure 5 less necessary. In the caption of Table 2 we will state that Figure 5 shows the results of the mixed-effects models.

Reviewer comment: In Fig. 7 it is hard to read the confidence intervals due to overlapping the same colors.

Author response: We will make the confidence intervals more clearly in the next version of the manuscript.

Reviewer comment: There is serious cherry-picking in the 2019 graphs in fig. 8 and fig. 9. In 2019 only three plots were measured and then the authors select the one that support their conclusions the best to include in the paper. The supporting figures of the remaining 2 plots that were measured in 2019 don't show the same strong results as the ones that are included in the main paper.

Author response: We decided to move Figure 8 to the Appendix alongside the time series of the other two plots of 2019. The purpose of Figure 9 was to show that the difference in microclimatic conditions between the lichen and shrub plots is larger for clear, sunny days than for cold, cloudy days. Therefore, we chose a distinct warm,

sunny day and a distinct cold, cloudy day, as describe in the methods (Line 173-175).

Reviewer comment: Generally, it is a good study with only 13 plots over two years, and only a few consecutive days of measurements. 2018 was a very warm and dry year. Their plot selection could have introduced bias. The authors should make their claims according to the limits of their experiment. The same group of authors have published multiple papers focusing on slightly different aspects of this same site and experiment, further decreasing the novelty of the results presented here. My suggestion would be to extend the study by revisiting the sites in a different year, or performing the same experiment at a different site. The results would be very much strengthened by such an extension of the study and the scientific community would benefit from a more thorough investigation of this interesting experiment.

Author response: We thank the Reviewer that he/she acknowledges our study. The Reviewer is concerned about the amount of plots and years on which we base our conclusions. We would argue that our analysis shows that there is no sign of a significant bias, and that our dataset robustly supports our conclusion. Indeed, similar studies measuring with radiometers and soil heat flux systems (e.g. Blok et al. 2010; Juszak et al. 2016) have often measurements during one year and less plots. Studies with radiometers and soil heat flux systems like these are often limited by the amount of sensors. Therefore, it is hard to build proper replication into a sampling design. We would like to have measurements on more plots, however, this would also mean that we need to change plots more often, which would lead to less measurement days. We would also like to have more measurement days, but this would mean that we would have less plots. Therefore, we chose a hybrid solution with measuring one year multiple plots and less consecutive measurement days (2018) and the other year measuring less plots and more consecutive days (2019). Since our plot selection in 2019 was not random and the sample design was different for the field season of 2019 than for the field season of 2018, we chose to use only the plots of the field season of 2018 for the main analysis and use the plots of the field season of 2019 to support the results of

the measurements of the field season of 2018. This means that we did not introduce bias in our main analysis due to plot selection. Indeed, 2018 was a warm and dry year and this was a reason to report the measurements of the field season of 2019 next to the analysis of the field data of 2018. As mentioned before, we will put emphasis on the limitations of our experiment in the next version of the manuscript. Moreover, we will elaborate on the consequences of the warm and dry field season of 2018 on our results. The Reviewer mentions further that we published a paper on a slightly different aspect of the same experiment. However, in this other paper (Aartsma et al., 2020) we measured the difference in albedo between lichen heaths and shrubs while in the current manuscript we study the effect of this difference in albedo on other micro-climatic variables. Therefore, we think that this current manuscript brings sufficiently novel results to be published on its own. In the next version of the manuscript, we will introduce our findings and link the results of the previous paper (Aartsma et al., 2020) more thoroughly to the study of the current manuscript.

Reviewer comment: General comments: l.32 add a Norwegian or at least Scandinavian example?

Author response: We will add a Scandinavian example in the next version.

Reviewer comment: l.35 what about increased competition due to e.g. grazing?

Author response: Yes, grazing and trampling of lichens could lead to a decrease in abundance of lichens. We will add this in the next version.

Reviewer comment: l.42 does vegetation composition indicate the vegetation community, abundance, diversity?

Author response: In the sentence following line 42, we give examples on how the shrub increase alters the vegetation composition. We mention that a negative relationship between shrub and lichen occurrence is reported by multiple studies (line 43-44). We also mention that shrub patches have a lower lichen abundance and diversity than

areas without shrubs (line 43-44).

Reviewer comment: l.47 what other distinct characteristics other than albedo?

Author response: We will add more examples of the distinct vegetation characteristics between lichens and shrubs, for example their difference in vegetation height and the strong insulating capacity of lichens.

Reviewer comment: l.81 expand on insulating properties of lichens? Include differences between e.g. crustose and foliose lichens and indicate which type of lichen species were dominant in their plots?

Author response: We will expand on the insulating properties of lichens in line 47, in which we discuss the distinct characteristics of lichens and shrubs.

Reviewer comment: l. 108 'similar way, similar positions', explain how exactly?

Author response: We will explain this more accurately in the next version.

Reviewer comment: l.111 30cm above canopy. So much higher for shrubs with a much higher canopy? Consider other effects such a surface roughness and turbulence affecting this.

Author response: The radiometers were placed 30 cm above the vegetation canopy. Since these radiometers measured only the radiation terms (shortwave and longwave radiation), we think that these measurements are not affected by the surface roughness and turbulence. This would be different if we would measure the latent and sensible heat flux on this height, however, we did not measure these fluxes directly.

Reviewer comment: Tbl.2 move to supplementary information.

Author response: As proposed by Reviewer 1, we will add the slope and error of the fixed effects in Table 2. This provides more useful information for the reader. Since the table will increase in size when the slope and error of the fixed effects are added, we choose to report the output of the mixed models for the daily averages/totals in

the main text and move the output of the mixed models for the daytime and nighttime averages/totals to the Appendix.

References used by the author:

Aartsma, P., Asplund, J., Odland, A., Reinhardt, S., & Renssen, H. (2020). Surface albedo of alpine lichen heaths and shrub vegetation. Arctic, Antarctic, and Alpine Research, 52(1), 312-322.

Blok, D., Heijmans, M., Schaepman-Strub, G., van Ruijven, J., Parmentier, F., Maximov, T., & Berendse, F. (2011). The cooling capacity of mosses: controls on water and energy fluxes in a Siberian tundra site. Ecosystems, 14(7), 1055-1065.

Juszak, I., Eugster, W., Heijmans, M. M. P. D., & Schaepman-Strub, G. (2016). Contrasting radiation and soil heat fluxes in Arctic shrub and wet sedge tundra. Biogeosciences, 13(13), 4049.

Nakagawa, S., & Schielzeth, H. (2013). A general and simple method for obtaining R2 from generalized linear mixed-effects models. Methods in ecology and evolution, 4(2), 133-142.

---

## Author Comment (AC3) · 18 Dec 2020

We thank Reviewer 3 for his/her time to review our manuscript and for his/her valuable comments. Below are our answers on his/her comments.

Reviewer comment: This study examines differences in surface energy partitioning and soil microclimate between lichen- and shrub-dominated vegetation in southern Norway using paired measurements made with a set of mobile instrument platforms. The authors find that the lower albedo of shrub canopies leads to higher atmospheric heating, but lichen mats have greater soil heat fluxes and temperatures despite lower net radiation. The latter is attributed shrub canopy shading and a thicker litter layer with

lower thermal conductivity. The results provide important context for understanding how shrub expansion will affect microclimate when shrubs replace lichens. The paper is well written, interesting, and I enjoyed reading it. There are several improvements that could help to strengthen the paper before it is considered further for publication.

Author response: We thank the Reviewer for his/her positive words on the writing style and interest in our paper. Moreover, we thank him/her for his/her advices on how to strengthen the paper.

Reviewer comment: While I don't think there are any technical problems with how the methods were applied, I do find it curious that different approaches were used to analyze the data from different years, and that the data weren't aggregated. Why not use data from 2018 and 2019 in the mixed models to examine differences in microclimate (i.e. in Table 2)?

Author response: Initially, the sample design of this study contained only measurements of the field season of 2018. However, 2018 was an unusually dry and warm year and therefore we decided to include measurements of the field season of 2019, since they support our findings of the field season of 2018 under less extreme conditions. We decided to use only the measurements of the field season of 2018 for the mixed models, because there are quite some differences in the sample design between the two years. For example, the plot pairs of the field season of 2018 consist of two measurement days, while plot pairs of the field season of 2019 consist of six measurement days per plot pair. Moreover, the location of the plots of the field season of 2018 were drawn randomly with ArcMap, while the locations of the plots of the field season of 2019 were chosen subjectively. Therefore the inclusion of the plots of 2019 in the mixed models might introduce selection bias. Therefore, we chose to do the main analysis on the data of 2018 only and not to include the data of 2019. We see that the difference in sample design between both years and the way of analysis might lead to confusion by the reader. Therefore, we will elaborate more on the reasons for the difference in sample design for both years and our way of analysis in the next version
of the manuscript.

Reviewer comment: A box and whisker plot showing mean microclimate by vegetation type would be more informative than the individual daily averages shown in Figure 4. Something like Figure 3, but instead showing net radiation, soil heat flux, soil temperature, etc. .

Author response: The Reviewer proposes to replace Figure 4 of the manuscript for Figure 1 of this answer (see below). Although we see the advantages of plotting the difference in microclimatic conditions between the lichen and shrub plots as proposed by the Reviewer, we still think that the current Figure 4 is more informative for the reader. The current Figure 4 adds important information on the timing, as it shows when and how the microclimatic conditions differed between the lichen and paired shrub plots during the field season. Moreover, the current Figure 4 gives information to the reader on how the sample design was set up during the field season of 2018. For example, the reader can see that we measured two days per plot pair and subsequently change to another plot pair. Therefore, we propose to keep Figure 4 in the manuscript. However, if the Reviewer and/or the Editor find the figure above relevant as well, we will add it to the new version of the manuscript.

Reviewer comment: I also wonder whether it would be more appropriate to model microclimate conditions as a function of incident shortwave radiation, rather than temperature, since this likely affects soil temperature and heat flux more so than air temperature (e.g. L163-165)?

Author response: We have modelled the daily average soil temperature and daily total soil heat flux as a function of the incident shortwave radiation. Subsequently we have determined the marginal r-squared for both models as proposed by Nakagawa & Schielzeth (2013). The marginal r-squared is the variation that is explained by the fixed effects. It turned out that the marginal r-squared was higher when we modelled the microclimatic variables as a function of air temperature ($R^2 = 0.58$ for daily total

soil heat flux and R2 = 0.43 for daily average soil temperature) than as a function of incident shortwave radiation (R2 = 0.40 for daily total soil heat flux and R2 = 0.14 for daily average soil temperature). Moreover, we construct models for the average soil temperature and the total soil heat flux separately for nighttime, when incident short-wave radiation is absent. Therefore, we opt to analyze the microclimate as a function of air temperature rather than as a function of incident shortwave radiation.

Reviewer comment: Related, are Figure 5 & 7 showing results of the mixed effects models?

Author response: Yes, Figure 5 shows the results of the daily averages/daily totals and Figure 7 shows the results of the daytime and nighttime averages/totals. We will make this clear to the reader by adding this information in the captions of Figures 5 & 7 in the next version of the manuscript.

Reviewer comment: Also I recommend that the authors consult Loranty et al 2018 published in PLOS One. This paper examines differences in soil temperature, thermal conductivity, surface temperature, and ET between lichen- and shrub-dominated vegetation patches in Siberia. It reaches many of the same conclusions presented in this manuscript, and would provide useful context in the introduction and discussion. Loranty, M.M., Berner, L.T., Taber, E.D., Kropp, H., Natali, S.M., Alexander, H.D., Davydov, S.P. and Zimov, N.S., 2018. Understory vegetation mediates permafrost active layer dynamics and carbon dioxide fluxes in open-canopy larch forests of northeastern Siberia. Plos one, 13(3), p.e0194014.

Author response: We thank the Reviewer for sharing this article with us. We were not aware its existence and see its value for our manuscript. Therefore, we will implement the findings of the paper of Loranty et al. in the introduction and discussion of the next version of our manuscript. Moreover, it shows that the results of our study are consistent with studies at other alpine/Arctic areas.

Reviewer comment: Minor comments: L1: Does your study really address the decline

of lichen heaths? The results certainly have implications in this context, but it seems more like a comparison between lichen heaths and shrubs. Something to that effect would be more appropriate in the title.

Author response: We understand the point of the Reviewer and will change the title to the following: "Microclimatic comparison of lichen heaths and shrubs: shrubification generates atmospheric heating but subsurface cooling during the growing season."

Reviewer comment: L45-50: This seems like it refers to another study that uses the same data presented in this manuscript. It would be appropriate to note that.

Author response: We only use to a small extent the same data as for the study of Aartsma et al. (2020). In the earlier study we quantified the difference in albedo between lichen and shrub plots, and in the current manuscript we study the effect of this difference in albedo on other microclimatic variables. As proposed, we will link the findings of the earlier paper (Aartsma et al. 2020) more clearly to this study in the next version of the manuscript.

Reviewer comment: L80: It should be made clear here that the authors know albedo is higher for lichen as a result of previous analyses from this data set.

Author response: We will make this clear in the next version.

Reviewer comment: L85: Is there permafrost at this site?

Author response: No, there is no permafrost at this site. We will mention that in the section "Study Area" in the next version.

Reviewer comment: L216: The wording "As for the net radiation" is a little confusing for me.

Author response: We will rephrase this wording in the next version.

Reviewer comment: L229: It is worth noting here that alpine tundra and lichen mats will also have different albedos, and the while lichen does have high albedo and can

be somewhat abundant, it is not broadly representative of alpine tundra.

Author response: We understand the point that is made by the Reviewer and will give some more context on the representability of lichen heaths in alpine areas in the next version of this manuscript.

Reviewer comment: L240: See Loranty et al 2018 in PLOS One for surface temperature and ET measurements of lichens vs shrubs.

Author response: We will use the findings by Loranty et al. to provide some more context on our results.

Reviewer comment: L303: ET would affect the canopy temperature more than that ground temperature, and as noted on line 285 the latter likely has more impact on soil temperature. However, higher ET shrubs may cool canopy temperatures, meaning less LW emitted from the canopy, and LW enhancement by canopies can affect the energy balance at the ground surface (e.g. Todt et al, 2018; Wake et al, 2017). This hasn't been shown in shrub tundra, but might be worth considering here.

Author response: We thank the Reviewer for pointing to this mechanism and the belonging articles. We will mention the mechanism in the new version of the manuscript.

Reviewer comment: L361-365: This section could be developed a bit more. It would be worth discussing how pervasive lichens are across alpine and arctic regions more generally. What types of modeling studies might your measurement help to inform, ecosystem or global scale studies, are there any example citations? Which measurements specifically might be useful for modeling?

Author response: We will develop this part a bit more and elaborate how our findings can be used for further (modelling) studies. Lichen heaths are often not incorporated in land surface models or they are clumped together with mosses, despite their different characteristics (Porada et al., 2016; Stoy et al., 2012; Wullschleger et al., 2014). Our study can help to develop a first version of lichen heaths as a separate plant functional

type in land surface models.

Reviewer comment: Figure 7: Why aren't the points included here, but included in Figure 5?

Author response: We decided not to include the points in Figure 7, since the figure would become rather messy when we would include the actual measurements of the microclimatic variables for both daytime and nighttime. We understand that it might lead to confusion for the reader that in Figure 5 the actual measurements are indicated and in Figure 7 not. Therefore, we decided to remove the points from Figure 5.

Reviewer comment: Figure 9: The abbreviations ST, SHF, etc. should be used consistently throughout the figures/manuscript.

Author response: We will take care that these abbreviation are used consistently throughout the next version of the manuscript.

Reviewer comment: References: Todt, M., Rutter, N., Fletcher, C.G., Wake, L.M., Bartlett, P.A., Jonas, T., Kropp, H., Loranty, M.M. and Webster, C., 2018. Simulation of longwave enhancement in boreal and montane forests. Journal of Geophysical Research: Atmospheres, 123(24), pp.13-731.

Webster, C., Rutter, N. and Jonas, T., 2017. Improving representation of canopy temperatures for modeling subcanopy incoming longwave radiation to the snow surface. Journal of Geophysical Research: Atmospheres, 122(17), pp.9154-9172.

Additional references by authors:

Aartsma, P., Asplund, J., Odland, A., Reinhardt, S., & Renssen, H. (2020). Surface albedo of alpine lichen heaths and shrub vegetation. Arctic, Antarctic, and Alpine Research, 52(1), 312-322.

Nakagawa, S., & Schielzeth, H. (2013). A general and simple method for obtaining R2 from generalized linear mixed‐effects models. Methods in ecology and evolution,

4(2), 133-142.

Porada, P., Ekici, A., & Beer, C. (2016). Effects of bryophyte and lichen cover on permafrost soil temperature at large scale. The Cryosphere, 10(5), 2291.

Stoy, P. C., Street, L. E., Johnson, A. V., Prieto-Blanco, A., & Ewing, S. A. (2012). Temperature, heat flux, and reflectance of common subarctic mosses and lichens under field conditions: might changes to community composition impact climate-relevant surface fluxes? Arctic, Antarctic, and Alpine Research, 44(4), 500-508.

Wullschleger, S. D., Epstein, H. E., Box, E. O., Euskirchen, E. S., Goswami, S., Iversen, C. M., Kattge, J., Norby, R. J., van Bodegom, P. M., & Xu, X. (2014). Plant functional types in Earth system models: past experiences and future directions for application of dynamic vegetation models in high-latitude ecosystems. Annals of Botany, 114(1), 1-16.
* * *
[Figure]

[Figure]

**Fig. 1.** The figure that should replace Figure 4 of the manuscript as proposed by the Reviewer.

---

## Author Response (AR1)

**Reviewer 1**

We thank Reviewer 1 for his/her time to review our manuscript and for his/her valuable comments. Below are our answers on his/her comments. The line numbers in our answers refer to the manuscript with track changes.

Reviewer comment:

This paper investigates the effect of lichen cover on alpine tundra microclimates, relative to increased shrub cover growth. The authors use two years worth of vegetation and microclimate data collected in Alpine Norway and find that a decline in lichens and increase in shrub cover lead to an overall atmospheric heating effect. I think this is a really well framed research gap highlighting an understudied and potentially very important research gap. While the data only come from one site, I think the topic is in the scope of Biogeosciences, as it has potential tundra-wide implications. I think the paper is overall clearly written, with a thorough methodology, and interesting discussion. However, I propose that this paper be reconsidered after major revisions because I feel like more work could be done to frame the research in the context of ecosystem-wide change, and I think more work could be done to frame hypotheses from the beginning of the paper.

Author response:

We thank the Reviewer for his/her positive words concerning the importance of our study and the writing style of the paper. Below we reply in detail to his/her comments concerning the framing of the hypotheses and the framing of our research in the context of ecosystem-wide change.

Reviewer comment:

The introduction and literature review is concise and persuasive, but I feel as though you could state clear hypotheses at the end of this section to better frame the rest of the paper. You have nice comprehensive microclimate measurements, and your methods section is replicable and thorough. I think your results section works well although I have provided some comments on the figures. Finally, I'd like to see more discussion of the implications of these findings for other alpine and Arctic tundra sites - do you expect to see similar trends across the biome? What significance do you think this has for the ecosystem as a whole? Do you expect to see similar trends in the Arctic tundra too, or just Alpine sites - and what is your justification for this? Finally, how much do you think shrub species matter in explaining the variation in your data? Are all tundra shrubs out-competing all lichens? I think this could be an interesting line of enquiry. I'm looking forward to reading this paper again!

Author response:

We thank the Reviewer for his/her positive feedback concerning the introduction, methods and results section. In the new version of the manuscript, we have listed the hypotheses at the end of the introduction in order to frame the rest of the paper (line 90-93). We have also discussed more extensively the more general implications of our findings for other alpine and Arctic areas (line 396-409). We see that this is necessary since our measurements were conducted on one field site. To elaborate on the question if we expect similar trends across the biome, we included more studies that

have been measuring microclimatic conditions below lichens and shrubs in alpine and Arctic areas before (line 399-401). Moreover, we used articles that studied the implications of shrubification on alpine and Arctic areas, to discuss if we expect different results between different tundra shrubs species and shrub characteristics (e.g. shrub height, shrub density) concerning the variation in microclimate between lichens and shrubs (line 403-409). We agree that this additional discussion is of relevance for our manuscript.

Reviewer comment:
- generally, you should capitalise 'Arctic'

Author response:
We capitalized "Arctic" throughout the manuscript.

Reviewer comment:
-21: "we hypothesise" - reword. The placement of this phrase in the abstract makes it seem like a hypothesis you set at the beginning relating to this project as opposed to a theory you generated based on your results

Author response:
We see the point that the Reviewer makes and replaced "hypothesize" with "predict".

Reviewer comment:
- 33: "this region" - I understand from context, but you could make it clearer which region you mean in this sentence (i.e. are you still talking about the Western Canadian Arctic, or Alpine Norway again?)

Author response:
We have made this clearer with replacing "this region" with "the western Canadian Arctic" (line 36).

Reviewer comment:
- 36: not clear from this sentence if lichen is declining purely because of competition with shrubs, or if they aren't coping physiologically with a rapidly warming temperature

Author response:
We have made this clear by adding that lichens will decline due to an increase in height and density of shrubs and graminoids, as indicated by the references within this sentence (line 42-43)

Reviewer comment:
- 61: I'd call it "macroclimate" instead of "large climate"

Author response:
We replaced "large-scale climate" with "macroclimate" throughout the manuscript.

Reviewer comment:
- Figure 2:: could you add the letter labels to the photography too? Its currently quite hard to envision.

Author response:
We added the letter labels in the photography of Figure 2.

Reviewer comment:
- 161: be explicit here, what is the interaction and why did you include it? We get the interaction later on in the tables but I feel you could spell it out here.

Author response:
We elaborated more on the interaction term in line 184-186.

Reviewer comment:
- Tab 2: maybe you should also add a column saying how much variation in % is explained by your random effects? Also you should add the slope/estimate and error in this table, as the F and P values are not very informative on their own.

Author response:
We added the slope and error in Table 2 and added a column saying the marginal and conditional r-squared for each model. The marginal r-squared is the variation explained by the fixed effects and the conditional r-squared is the variation explained by the entire model. Since Table 2 was rather large when we added this information for the daily, daytime and nighttime analysis, we chose to report only a table with the analysis of the daily averages/totals in the main text and placed the table with the analysis of the daytime and nighttime averages/totals in the Appendix.

Reviewer comment:
- Figure 4: a nice clear figure. Move the (d) label up a bit, it looks a bit cluttered at the moment.

Author response:
We moved the (d) label a bit up.

Reviewer comment:
- Figure 5: looks great but I'd recommend changing the colours from red to blue to something less indicitive of temperature. Currently it suggests lichens = hot, shrubs = cold.

Author response:
We changed the colors in all the figures from red and blue into light-green (for lichens) and dark-green (for shrubs).

Reviewer comment:
- Figure 8: useful site information, but maybe move to supplementary materials? You have a lot of figures and I feel like this one can be moved.

Author response:
We moved Figure 8 to the Appendix.

Reviewer comment:
- 298: "tundra plots" vs "shrub plots" = this is confusing because tundra can be very shrubby. Maybe rename these descriptors to make this more explicit?

Author response:
We see the point made by the Reviewer and changed the term "tundra plots" into "open tundra plots" (line 319). The plots in this line refers to the study of Myers-Smith and Hik (2013), who refer to them as "open plots" in their paper.

**Reviewer 2**

We thank Reviewer 2 for his/her time to review our manuscript and for his/her valuable comments. Below are our answers on his/her comments. The line numbers in our answers refer to the manuscript with track changes.

Reviewer comment:

This paper investigates the differences between lichen and shrub cover for alpine tundra microclimates. The authors measured vegetation and microclimate data at a site in Norway in two consecutive years. The research gap that they are aiming to investigate is well-framed and interesting for the field of tundra ecology in a wider scope. The article is well-written, especially the discussion and methods are overall clear. The main strength of the paper in my view is that it introduces meticulous measurements on an understudied aspect of this field. However, I propose major revisions to this paper before acceptance. The authors should reframe the conclusions of their study in accordance to the size and caveats of their experiment and improve on their analysis of the results.

Author response:

We thank the Reviewer for his/her positive words about the framing of the research gap and the writing style of the manuscript. We understand his/her point concerning the size and caveats of our experiment and elaborated on this in the new version of the manuscript. We have in particular framed our experiment, which is based on one study area, in the context of ecosystem-wide change over the Northern Hemisphere (line 396-409). Below we will expand on our additional thoughts about the size of our experiment and the analysis of the results by addressing the more specific comments.

Reviewer comment:

The paired plot set-up, and the plot selection not being random, risks the introduction of a selection bias. Despite their thorough methodology it remains unclear as to how their plots where actually selected. Was selection bias prevented through radial/cardinal walking?

Author response:

For the plots of 2018, we selected randomly 10 locations in the study area using ArcMap. Subsequently, we selected the lichen plots within a radius of 50 meters around each random location that fitted the criteria in Table 1. In case we found multiple lichen plots that fitted these criteria, we selected the lichen plots with the highest percentage of lichens. Subsequently, we selected the shrub plots within a radius of 50 meters around each selected lichen plot that fitted the criteria in Table 1. In case we found multiple shrub plots that fitted these criteria, we selected the shrub plot with the highest percentage of shrubs. The location of the plots of 2019 were selected subjectively, but still fulfilled the criteria of Table 1. Since the location of the plots of 2019 were selected subjectively, we decided to include only the measurements of 2018 in the mixed effects models and in the Wilcoxon test and therefore bias was prevented in these analyses.

We agree that we did not specify the plot selection adequately in the text and explained this more detailed in the new version (line 114-118).

Reviewer comment:
How was a plot determined to be on a ridgetop, and how was a slope <10% determined?

Author response:
The ridgetops in our study area are well distinct from the midslopes and the snowbeds (see Appendix A of the manuscript for a picture of the study site). The ridgetops were determined based on their vegetation characteristics and the low or absent slope angle. We obtained the slope of each plot with a clinometer. We have specified this in Table 1.

Reviewer comment:
Why was this area picked in general, does it represent the landscape that they want to study in a particularly good way?

Author response:
We selected this area since it has a high abundance of lichens and the landscape is representative for an alpine landscape in Scandinavia. We elaborated on this in the section "Study area" (line 102-103).

Reviewer comment:
The authors successfully performed very high-detail measurements for both radiation and soil parameters. The measurements where however always on different days for different plots, and there was no revisiting of same plots, not even in 2019 when they revisited the study site. This limited amount of measuring days, measured plots and no revisiting of plots increases the risk of the results being affected by unaccounted variability. 2018 for instance was a very warm and dry year that could influence many processes in the ecosystem. Please expand on the potential consequences of these limitations and why the experiment design was changed in 2019.

Author response:
The initial sample design for this study contained only measurement of one field season (2018). However, 2018 turned out to be a warm and dry year and therefore we chose to include measurements of 2019 in this manuscript to show that the measurements of 2019 show a similar pattern (lichen plots having a higher soil temperature and soil heat flux than shrub plots during warm days, despite having a lower net radiation). Unfortunately, we could not revisit the same plots of the field season of 2018, since the vegetation was disturbed after taking soil samples, and this affects the measurements (especially radiation) substantially. The reason for a change in experimental design was that we wanted to measure the plots for a longer period to gain insights in the variability in the microclimatic conditions over time.

In the next version of this manuscript, we elaborated more on the consequences of the warm and dry field season for our study in the discussion (line 397-402). Moreover, we gave our reasons for the difference in experimental design between the two field seasons (line 149-150).

Reviewer comment:
The authors perform their statistics based on a sample size of 13, Wilcoxon is however not a parametrized method, and generally not suitable for small sample sizes. Significance of results based on these tests can't be concluded with so much certainty as the authors claim. Consider trying a paired parametric test. Be aware of confounding variables in the microclimate.

Author response:
As suggested by the reviewer, we repeated the analysis with the paired parametric t-test and the paired permutation test. Both tests draw the same conclusion as the Wilcoxon test. However, in our view, the Wilcoxon test is the appropriate test to use here. The parametric t-test has the underlying assumption that the data has a normal distribution. Since we cannot test this assumption due to the low sample size, we decided to use the more conservative Wilcoxon test. Therefore, we still opt to report the results of the Wilcoxon test. Note: we analyzed the difference in the canopy and soil variables between the lichen and shrub plots (Fig. 3) for the plots of the field season of 2018 only, and therefore the sample size consists of 10 plot pairs. We made this clear in the new version by adding "of 2018" and "n = 10 for each boxplot" in the caption of Figure 3.

Reviewer comment:
Comments on figures: the amount of figures in the paper could be decreased.

Author response:
We moved Figure 8 to the Appendix.

Reviewer comment:
Fig. 3 needs better axis scaling to highlight their results.

Author response:
We changed the scaling of the y-axis of this figure.

Reviewer comment:
Fig. 5 could benefit from an r2 to indicate the strength of relationship.

Author response:
We determined the marginal and conditional r2 of each linear mixed-effects model as proposed by Nakagawa & Schielzeth (2013) and indicated these r-squares in Table 2. The marginal r-squared is the variation explained by the fixed effects and the conditional r-squared is the variation explained by the entire model. We chose to report these values in Table 2, since Reviewer 1 requested this information in this table. Moreover, we decided to remove the actual measurements from Figure 5 to be consistent with Figure 7, which made the indication of the r2 in Figure 5 less necessary. In the caption of Table 2 we state now that Figure 5 shows the results of the mixed-effects models.

Reviewer comment:
In Fig. 7 it is hard to read the confidence intervals due to overlapping the same colors.

Author response:
We made the confidence intervals clearer by using different colors for the daytime and the nighttime analyses.

Reviewer comment:
There is serious cherry-picking in the 2019 graphs in fig. 8 and fig. 9. In 2019 only three plots were measured and then the authors select the one that support their conclusions the best to include in the paper. The supporting figures of the remaining 2 plots that were measured in 2019 don't show the same strong results as the ones that are included in the main paper.

Author response:
We decided to move Figure 8 to the Appendix (Fig. E2) alongside the time series of the other two plots of 2019 (Fig. E1 and Fig. E3).
The purpose of the former Figure 9 (current Fig. 8) was to show that the difference in microclimatic conditions between the lichen and shrub plots is larger for clear, sunny days than for cold, cloudy days. Therefore, we chose a distinct warm, sunny day and a distinct cold, cloudy day, as described in the methods (line 195-198).

Reviewer comment:
Generally, it is a good study with only 13 plots over two years, and only a few consecutive days of measurements. 2018 was a very warm and dry year. Their plot selection could have introduced bias. The authors should make their claims according to the limits of their experiment. The same group of authors have published multiple papers focusing on slightly different aspects of this same site and experiment, further decreasing the novelty of the results presented here. My suggestion would be to extend the study by revisiting the sites in a different year, or performing the same experiment at a different site. The results would be very much strengthened by such an extension of the study and the scientific community would benefit from a more thorough investigation of this interesting experiment.

Author response:
We thank the Reviewer that he/she acknowledges our study. The Reviewer is concerned about the amount of plots and years on which we base our conclusions. We would argue that our analysis shows that there is no sign of a significant bias, and that our dataset robustly supports our conclusion. Indeed, similar studies measuring with radiometers and soil heat flux systems (e.g. Blok et al. 2010; Juszak et al. 2016) have often measurements during one year and less plots. Studies with radiometers and soil heat flux systems like these are often limited by the number of sensors. Therefore, it is hard to build proper replication into a sampling design. We would like to have measurements on more plots, however, this would also mean that we need to change plots more often, which would lead to less measurement days. We would also like to have more measurement days, but this would mean that we would have less plots. Therefore, we chose a hybrid solution with measuring one year multiple plots and less consecutive measurement days (2018) and the other year measuring less plots and more

consecutive days (2019). Since our plot selection in 2019 was not random and the sample design was different for the field season of 2019 than for the field season of 2018, we chose to use only the plots of the field season of 2018 for the main analysis and use the plots of the field season of 2019 to support the results of the measurements of the field season of 2018. This means that we did not introduce bias in our main analysis due to plot selection. Indeed, 2018 was a warm and dry year and this was a reason to report the measurements of the field season of 2019 next to the analysis of the field data of 2018. As mentioned before, we have put emphasis on the limitations of our experiment in the next version of the manuscript and elaborated on the consequences of the warm and dry field season of 2018 on our results (line 396-409).

The Reviewer mentioned further that we published a paper on a slightly different aspect of the same experiment. However, in this other paper (Aartsma et al., 2020) we measured the difference in albedo between lichen heaths and shrubs while in the current manuscript we study the effect of this difference in albedo on other microclimatic variables. Therefore, we think that this current manuscript brings sufficiently novel results and new insights to be published on its own. In the new version of the manuscript, we introduced our findings and link the results of the previous paper (Aartsma et al., 2020) more thoroughly to the study of the current manuscript (line 90-91).

Reviewer comment:
General comments:
l.32 add a Norwegian or at least Scandinavian example?

Author response:
We added a Scandinavian example (line 37-38).

Reviewer comment:
l.35 what about increased competition due to e.g. grazing?

Author response:
Yes, grazing and trampling of lichens could lead to a decrease in abundance of lichens. We added this in the new version (line 39).

Reviewer comment:
l.42 does vegetation composition indicate the vegetation community, abundance, diversity?

Author response:
In the sentence following this line, we give examples on how the shrub increase alters the vegetation composition. We mention that a negative relationship between shrub and lichen occurrence is reported by multiple studies (line 49-50). We also mention that shrub patches have a lower lichen abundance and diversity than areas without shrubs (line 51-52).

Reviewer comment:
l.47 what other distinct characteristics other than albedo?

Author response:
We added more examples of the distinct vegetation characteristics between lichens and shrubs (line 54-55)

Reviewer comment:
l.81 expand on insulating properties of lichens? Include differences between e.g. crustose and foliose lichens and indicate which type of lichen species were dominant in their plots?

Author response:
We expanded on the insulating properties of lichens in line 54-55, in which we discuss the distinct characteristics of lichens and shrubs.

Reviewer comment:
l. 108 'similar way, similar positions', explain how exactly?

Author response:
We explained this more accurately in the new version (line 123-124).

Reviewer comment:
l.111 30cm above canopy. So much higher for shrubs with a much higher canopy? Consider other effects such a surface roughness and turbulence affecting this.

Author response:
The radiometers were placed 30 cm above the vegetation canopy. Since these radiometers measured only the radiation terms (shortwave and longwave radiation), we think that these measurements are not affected by the surface roughness and turbulence. This would be different if we would measure the latent and sensible heat flux on this height, however, we did not measure these fluxes directly.

Reviewer comment:
Tbl.2 move to supplementary information.

Author response:
As proposed by Reviewer 1, we added the slope and error of the fixed effects and the marginal and conditional r-squared in Table 2. This provides more useful information for the reader. Since the table will increase in size when this information is added, we chose to report the output of the mixed models for the daily averages/totals in the main text and moved the output of the mixed models for the daytime and nighttime averages/totals to the Appendix.

References used by the author:

Aartsma, P., Asplund, J., Odland, A., Reinhardt, S., & Renssen, H. (2020). Surface albedo of alpine lichen heaths and shrub vegetation. Arctic, Antarctic, and Alpine Research, 52(1), 312-322.

Blok, D., Heijmans, M., Schaepman-Strub, G., van Ruijven, J., Parmentier, F., Maximov, T., & Berendse, F. (2011). The cooling capacity of mosses: controls on water and energy fluxes in a Siberian tundra site. Ecosystems, 14(7), 1055-1065.

Juszak, I., Eugster, W., Heijmans, M. M. P. D., & Schaepman-Strub, G. (2016). Contrasting radiation and soil heat fluxes in Arctic shrub and wet sedge tundra. Biogeosciences, 13(13), 4049.

Nakagawa, S., & Schielzeth, H. (2013). A general and simple method for obtaining R2 from generalized linear mixed-effects models. Methods in ecology and evolution, 4(2), 133-142.

**Reviewer 3**

We thank Reviewer 3 for his/her time to review our manuscript and for his/her valuable comments. Below are our answers on his/her comments. The line numbers in our answers refer to the manuscript with track changes.

Reviewer comment:
This study examines differences in surface energy partitioning and soil microclimate between lichen- and shrub-dominated vegetation in southern Norway using paired measurements made with a set of mobile instrument platforms. The authors find that the lower albedo of shrub canopies leads to higher atmospheric heating, but lichen mats have greater soil heat fluxes and temperatures despite lower net radiation. The latter is attributed shrub canopy shading and a thicker litter layer with lower thermal conductivity. The results provide important context for understanding how shrub expansion will affect microclimate when shrubs replace lichens. The paper is well written, interesting, and I enjoyed reading it. There are several improvements that could help to strengthen the paper before it is considered further for publication.

Author response:
We thank the Reviewer for his/her positive words on the writing style and interest in our paper. Moreover, we thank him/her for his/her advices on how to strengthen the paper.

Reviewer comment:
While I don't think there are any technical problems with how the methods were applied, I do find it curious that different approaches were used to analyze the data from different years, and that the data weren't aggregated. Why not use data from 2018 and 2019 in the mixed models to examine differences in microclimate (i.e. in Table 2)?

Author response:
Initially, the sample design of this study contained only measurements of the field season of 2018. However, 2018 was an unusually dry and warm year and therefore we decided to include measurements of the field season of 2019, since they support our findings of the field season of 2018 under less extreme conditions. We decided to use only the measurements of the field season of 2018 for the mixed models, because there are quite some differences in the sample design between the two years. For example, the plot pairs of the field season of 2018 consist of two measurement days, while plot pairs of the field season of 2019 consist of six measurement days per plot pair. Moreover, the location of the plots of the field season of 2018 were drawn randomly with ArcMap, while the locations of the plots of the field season of 2019 were chosen subjectively. Therefore the inclusion of the plots of 2019 in the mixed models might introduce selection bias. Therefore, we chose to do the main analysis on the data of 2018 only and not to include the data of 2019.
We see that the difference in sample design between both years and the way of analysis might lead to confusion by the reader. Therefore, we elaborated more on the reasons for the difference in sample design for both years (line 149-151) and our way of analysis (line 178-180).

Reviewer comment:

A box and whisker plot showing mean microclimate by vegetation type would be more informative than the individual daily averages shown in Figure 4. Something like Figure 3, but instead showing net radiation, soil heat flux, soil temperature, etc. .

Author response:

The Reviewer proposes to replace Figure 4 of the manuscript by Figure 1 of this answer (see below). Although we see the advantages of plotting the difference in microclimatic conditions between the lichen and shrub plots as proposed by the Reviewer, we still think that the current Figure 4 is more informative for the reader. The current Figure 4 adds important information on the timing, as it shows when and how the microclimatic conditions differed between the lichen and paired shrub plots during the field season. Moreover, the current Figure 4 gives information to the reader on how the sample design was set up during the field season of 2018. For example, the reader can see that we measured two days per plot pair and subsequently change to another plot pair. Therefore, we propose to keep Figure 4 in the manuscript. However, if the Reviewer and/or the Editor find the figure below relevant as well, we will add it to the new version of the manuscript.

[Figure]

Figure 1: The figure that should replace Figure 4 of the manuscript as proposed by the Reviewer.

Reviewer comment:

I also wonder whether it would be more appropriate to model microclimate conditions as a function of incident shortwave radiation, rather than temperature, since this likely affects soil temperature and heat flux more so than air temperature (e.g. L163-165)?

Author response:

As suggested, we have now also modelled the daily average soil temperature and daily total soil heat flux as a function of the incident shortwave radiation. Subsequently we have determined the marginal r-squared for both models as proposed by Nakagawa & Schielzeth (2013). The marginal r-squared is the variation that is explained by the fixed effects. It turned out that the marginal r-squared was higher when we modelled the microclimatic variables as a function of air temperature ($R^2 = 0.58$ for daily total soil heat flux and $R^2 = 0.43$ for daily average soil temperature) than as a function of incident shortwave radiation ($R^2 = 0.40$ for daily total soil heat flux and $R^2 = 0.14$ for daily average soil temperature). Moreover, we construct models for the average soil temperature and the total soil heat flux separately for nighttime, when incident shortwave radiation is absent. Therefore, we opt keep the analysis of the microclimate as a function of air temperature rather than as a function of incident shortwave radiation.

Reviewer comment:
Related, are Figure 5 & 7 showing results of the mixed effects models?

Author response:
Yes, Figure 5 shows the results of the daily averages/daily totals and Figure 7 shows the results of the daytime and nighttime averages/totals. We made this clear to the reader by adding this information in the captions of Figures 5 and 7. Moreover, we added in the captions of Table 2 and Appendix C that Figure 5 and 7 visualize the results of the mixed models.

Reviewer comment:

Also I recommend that the authors consult Loranty et al 2018 published in PLOS One. This paper examines differences in soil temperature, thermal conductivity, surface temperature, and ET between lichen- and shrub-dominated vegetation patches in Siberia. It reaches many of the same conclusions presented in this manuscript, and would provide useful context in the introduction and discussion. Loranty, M.M., Berner, L.T., Taber, E.D., Kropp, H., Natali, S.M., Alexander, H.D., Davydov, S.P. and Zimov, N.S., 2018. Understory vegetation mediates permafrost active layer dynamics and carbon dioxide fluxes in open-canopy larch forests of northeastern Siberia. Plos one, 13(3), p.e0194014.

Author response:
We thank the Reviewer for sharing this article with us. We were not aware of its existence and see its value for our manuscript. Therefore, we implemented the findings of this paper in the introduction (line 77-79) and discussion (line 264-267, line 275-277, and line 344-345). Moreover, it shows that the results of our study are consistent with studies at other alpine/Arctic areas.

Reviewer comment:
Minor comments: L1: Does your study really address the decline of lichen heaths? The results certainly have implications in this context, but it seems more like a comparison between lichen heaths and shrubs. Something to that effect would be more appropriate in the title.

Author response:
We understand the point of the Reviewer and changed the title to the following:
"Microclimatic comparison of lichen heaths and shrubs: shrubification generates atmospheric heating but subsurface cooling during the growing season."

Reviewer comment:
L45-50: This seems like it refers to another study that uses the same data presented in this manuscript. It would be appropriate to note that.

Author response:
We only use to a small extent the same data as for the study of Aartsma et al. (2020). In the earlier study we quantified the difference in albedo between lichen and shrub plots, and in the current manuscript we study the effect of this difference in albedo on other microclimatic variables. As proposed, we linked the findings of the earlier paper (Aartsma et al. 2020) more clearly to this study in the next version of the manuscript (line 90-91).

Reviewer comment:
L80: It should be made clear here that the authors know albedo is higher for lichen as a result of previous analyses from this data set.

Author response:
We made this clear in the new version (line 90-91).

Reviewer comment:
L85: Is there permafrost at this site?

Author response:
No, there is no permafrost at this site. We added this information in the section "Study Area" (line 107).

Reviewer comment:
L216: The wording "As for the net radiation" is a little confusing for me.

Author response:
We rephrased this wording in the new version (line 231).

Reviewer comment:
L229: It is worth noting here that alpine tundra and lichen mats will also have different albedos, and the while lichen does have high albedo and can be somewhat abundant, it is not broadly representative of alpine tundra.

Author response:
We understand the point that is made by the Reviewer and gave some more context on the representability of lichen heaths in alpine tundra (line 244-245).

Reviewer comment:
L240: See Loranty et al 2018 in PLOS One for surface temperature and ET measurements of lichens vs shrubs.

Author response:
We used the findings by Loranty et al. to provide some more context on our results (line 264-267).

Reviewer comment:
L303: ET would affect the canopy temperature more than that ground temperature, and as noted on line 285 the latter likely has more impact on soil temperature. However, higher ET shrubs may cool canopy temperatures, meaning less LW emitted from the canopy, and LW enhancement by canopies can affect the energy balance at the ground surface (e.g. Todt et al, 2018; Wake et al, 2017). This hasn't been shown in shrub tundra, but might be worth considering here.

Author response:
We thank the Reviewer for pointing to this mechanism and the belonging articles. We mention this mechanism in the new version (line 348-353).

Reviewer comment:
L361-365: This section could be developed a bit more. It would be worth discussing how pervasive lichens are across alpine and arctic regions more generally. What types of modeling studies might your measurement help to inform, ecosystem or global scale studies, are there any example citations? Which measurements specifically might be useful for modeling?

Author response:
We developed this part a bit more and elaborated how our findings can be used for further (modelling) studies (line 391-395)

Reviewer comment:
Figure 7: Why aren't the points included here, but included in Figure 5?

Author response:
We decided not to include the points in Figure 7, since the figure would become rather messy when we would include the actual measurements of the microclimatic variables for both daytime and nighttime. We understand that it might lead to confusion for the reader that in Figure 5 the actual measurements are indicated and in Figure 7 not. Therefore, we decided to remove the points from Figure 5.

Reviewer comment:
Figure 9: The abbreviations ST, SHF, etc. should be used consistently throughout the figures/manuscript.

Author response:
We took care that these abbreviation are used consistently throughout the figures/manuscript.

Reviewer comment:
References: Todt, M., Rutter, N., Fletcher, C.G., Wake, L.M., Bartlett, P.A., Jonas, T., Kropp, H., Loranty, M.M. and Webster, C., 2018. Simulation of longwave enhancement in boreal and montane forests. Journal of Geophysical Research: Atmospheres, 123(24), pp.13-731.

Webster, C., Rutter, N. and Jonas, T., 2017. Improving representation of canopy temperatures for modeling subcanopy incoming longwave radiation to the snow surface. Journal of Geophysical Research: Atmospheres, 122(17), pp.9154-9172.

Additional references used by the author:

Nakagawa, S., & Schielzeth, H. (2013). A general and simple method for obtaining R2 from generalized linear mixed-effects models. Methods in ecology and evolution, 4(2), 133-142.

**List of most relevant changes**

The following list includes the most relevant changes that have been made to the revised manuscript:

- The title of the manuscript was changed so it covers our study more precisely.
- Our hypotheses are listed at the end of the introduction in order to better frame the rest of the paper.
- We elaborated on the reasons for the difference in sample design between the two field seasons and on the reasons why we did not aggregate the measurements of both field seasons for the analyses.
- The slope and standard error of the fixed effect, and the marginal and conditional r-squares are added to Table 2. Since this table would become rather large, we moved the output of the daytime and nighttime analysis to the appendix.
- Figure 8 was moved to the appendix.
- We included the findings of the study by Loranty et al. (2018) in the introduction and discussion, since this study reported similar conclusions on the differences in microclimatic conditions between lichens and shrubs as understory vegetation of a Larch forest as our study.
- We discussed the representability of our study for other alpine or Arctic sites more elaborately.

---

## Author Response (AR2)

We thank the Reviewers for their time to review the revised manuscript and for their positive feedback. Below we reply on the technical corrections that were suggested by Reviewer 1.

Reviewer comment:
Figure 2 - would it be possible to add the black 'blobs' to the ABCD labels in the photograph as they are in the transect diagram? I think this would make the diagram slightly more readable.

Author response:
We have added the black dots in the photograph.

Reviewer comment:
Line 230 - Maybe a more informative subtitle than 'Day vs night'? Perhaps 'net radiation differences in daytime vs night'?

Author response:
We have changed the subtitle into "Microclimatic differences in daytime vs night".

Reviewer comment:
Figure 4 - would it be possible to make the lichen plot yellow slightly darker? I find it hard to make out the exact limit of these bars because its quite similar to the white background

Author response:
We have made the lichen plot yellow slightly darker in Figure 4. In order to be consistent with the color, we also made the lichen plot yellow slightly darker in Figure 6 and Figure D1.